# Multibranch Unsupervised Domain Adaptation Network for Cross Multidomain Orchard Area Segmentation

**Ming Liu [1], Dong Ren [1,*], Hang Sun [1] and Simon X. Yang [2]**

[1] College of Computer and Information Technology, China Three Gorges University, Yichang 443002, China
[2] School of Engineering, University of Guelph, Guelph, ON N1G 2W1, Canada
* Correspondence: dren@ctgu.edu.cn; Tel.: +86-139-9774-7675

**Abstract:** Although unsupervised domain adaptation (UDA) has been extensively studied in remote sensing image segmentation tasks, most UDA models are designed based on single-target domain settings. Large-scale remote sensing images often have multiple target domains in practical applications, and the simple extension of single-target UDA models to multiple target domains is unstable and costly. Multi-target unsupervised domain adaptation (MTUDA) is a more practical scenario that has great potential for solving the problem of crossing multiple domains in remote sensing images. However, existing MTUDA models neglect to learn and control the private features of the target domain, leading to missing information and negative migration. To solve these problems, this paper proposes a multibranch unsupervised domain adaptation network (MBUDA) for orchard area segmentation. The multibranch framework aligns multiple domain features, while preventing private features from interfering with training. We introduce multiple ancillary classifiers to help the model learn more robust latent target domain data representations. Additionally, we propose an adaptation enhanced learning strategy to reduce the distribution gaps further and enhance the adaptation effect. To evaluate the proposed method, this paper utilizes two settings with different numbers of target domains. On average, the proposed method achieves a high IoU gain of 7.47% over the baseline (single-target UDA), reducing costs and ensuring segmentation model performance in multiple target domains.

**Keywords:** semantic segmentation; multi-target unsupervised domain adaptation; remote sensing; orchard area

## 1. Introduction

Orchard estimation is significant for orchard management, yield estimation, and industrial development. In recent years, the recognition and interpretation of land cover types (such as trees and roads) in remote sensing images have attracted increasing research interest [1–3]. For example, semantic segmentation models have been applied to urban planning [4], green tide extraction [5], farmland segmentation [6], and other fields. These successful cases offer a research basis for cross multidomain orchard area segmentation. However, a model trained on a dataset acquired from several specific regions cannot be generalized to the other areas. The superior performance of semantic segmentation models largely depends on the supervision of massive data and the use of similar feature distributions [7,8]. Performance degradation occurs when the utilized training and testing data possess different feature distributions: this situation is called the domain shift problem [9–11]. Notably, remote sensing technology can easily acquire a wide variety of data, which are sensitive to time and region changes, and their acquisition is affected by the different types, seasons, and sensors [12,13]. Therefore, remote sensing images usually have multiple domain distributions, and creating labeled training sets for such a large number of remote sensing images is an impossible task. In this case, it is crucial to learn the potential features of other images (target domain) by utilizing one image with labels (source domain).

Fortunately, unsupervised domain adaptation (UDA) helps models work on new domains without additional annotation work, which helps to alleviate the labeling effort required to train segmentation models. This paper studies UDA models based on adversarial learning [14,15], which reduces domain gaps and learn domain-invariant features through adversarial learning. However, traditional UDA models [16–19] were designed for single-target settings, so they still cannot achieve the expected results on other target domains, as shown in Figure 1. Multi-target UDA (MTUDA) enables the utilized model to adapt to multiple target domains, which is more suitable for realistic applications. However, MTUDA faces more complex unlabeled data, making it more challenging to implement. Saporta et al. [20] introduced multi-discriminator framework to reduce both source-target and target–target domain gaps, and introduced multi-target knowledge transfer framework to optimize the learning process. Roy et al. [21] proposed curriculum graph co-teaching, which uses two different classifiers to aggregate the feature information of similar samples.

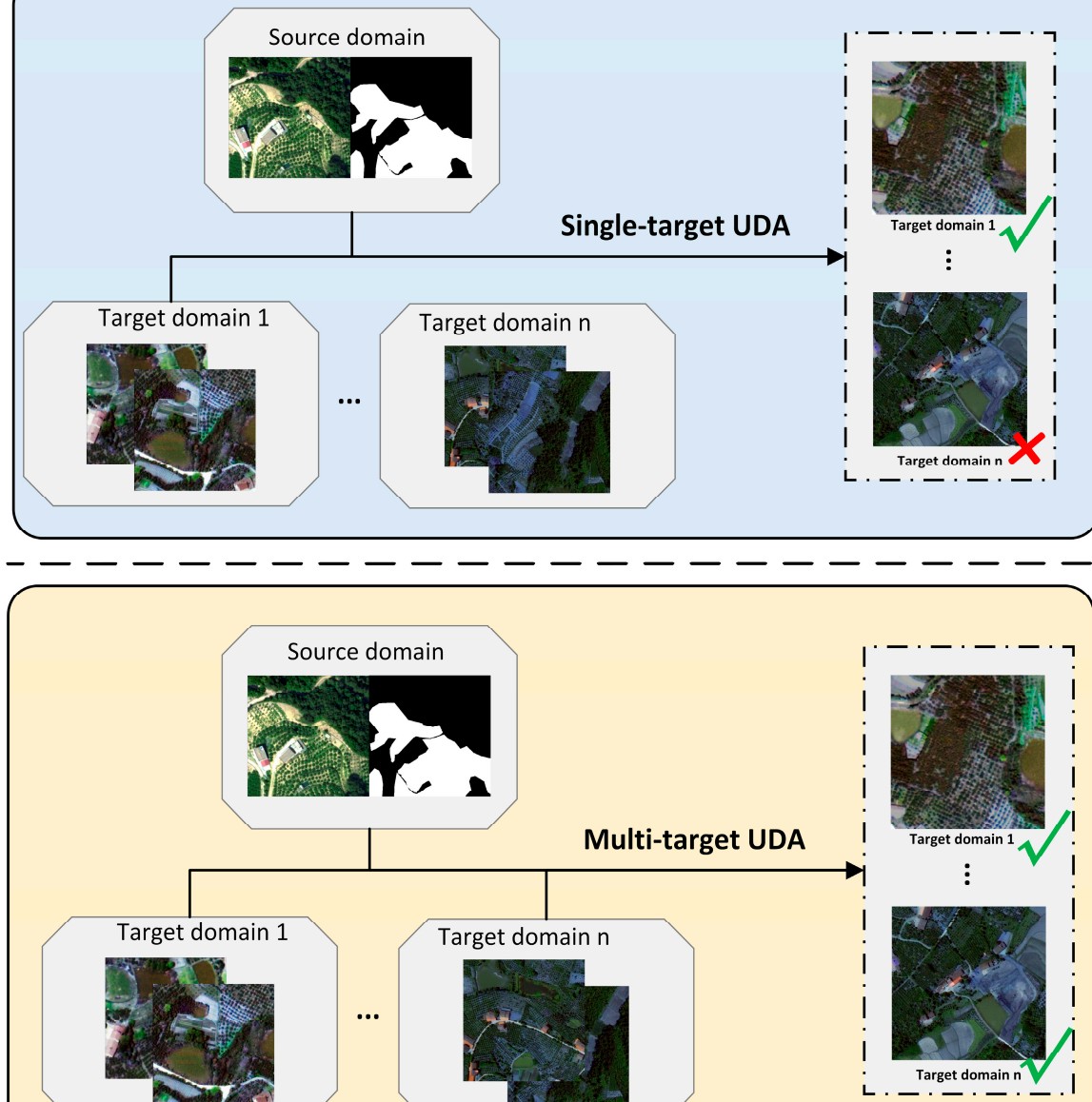

**Figure 1.** The different unsupervised domain adaptation scenarios in cross multidomain orchard area segmentation.

Similar to previous single-target UDA models [22–24], most MTUDA models [20,21] focus on the learning of domain-invariant features (correlated across the domains), while paying less attention to the private features (independent between the domains) of the target domain. This oversight leads to two problems; on the one hand, the private features of a particular target domain do not exist in the source domain, and forcing the alignment of these features may lead to negative migration, especially in the multi-target UDA task in which differences between domain features are more common; on the other hand, the private features are beneficial for the classifier when performing classification, and the neglect of private feature learning results in suboptimal performance for the trained segmentation model. These factors motivate us to incorporate private feature control and learning into the model training process to achieve better adaptation. In addition, since UDA needs to consider different distribution gaps, Pan et al. [25] discussed the effect of intra-domain gaps and achieved better results, but this method was designed based on single-target domain adaptation. MTUDA models also need to consider the target–target domain gaps, and how reducing different gaps is also a key to enhancing the adaptation effects of these models.

In this study, we propose multibranch UDA (MBUDA) for cross multidomain orchard area segmentation. Specifically, the multibranch framework separates domain invariant features and private features to prevent negative migration during training, and ensures independence between multiple target domains. Furthermore, this paper introduces multiple ancillary classifiers to encourage the model to learn domain-invariant features and private features so that a better latent representation of the target domain data can be obtained. To enhance the adaptation effect, this paper designs an adaptation enhanced learning strategy to reduce the target–target domain gaps and intra-domain gaps. The main contributions of this work are listed as follows:

(1)     This paper proposes a novel MTUDA network called MBUDA for cross multidomain orchard area segmentation; the designed multibranch structure and ancillary classifiers enable the segmentation model to learn the better feature representation of the target domains by learning and controlling the private features;

(2)     To further enhance the adaptation effect, an adaptation enhanced learning strategy is designed to refine the training process, which directly reduces the target–target gaps by aligning the features of target domain images with different confidence;

(3)     This paper designed various experiments to demonstrate the validity of the proposed methods, indicating that the proposed MBUDA method and adaptation enhanced learning strategy both achieve superior results to those of current approaches.

## 2. Materials and Methods

### 2.1. Related Work

#### 2.1.1. Unsupervised Domain Adaptation

Unsupervised Domain Adaptation is designed to learn a model based on a source domain that can generalize well to target domains with different feature distributions. The common methods of UDA can be divided into adversarial discriminative models, style transfer methods, and self-supervision methods. Adversarial discriminative models confuse the features of different domains and reduce the domain gaps through adversarial learning. Tsai et al. [26] proposed aligning the edge distributions in the output space. Subsequently, Vu et al. [27] proposed a depth-aware adaptation scheme that introduces additional depth-specific adaptation to guide the learning of the model. To further enhance the domain adaptation models, some researchers focused on the underlying structures among classes. Wang et al. [22] discussed the importance of class information and proposed a fine-grained discriminator that incorporates class information, while new domain labels were designed as new supervisory signals. Du et al. [16] used a progressive confidence strategy to independently adapt separated semantic features, which also reasonably utilizes the class information. For style transfer methods, scholars have used generative adversarial networks (GANs) [28] to reduce the differences between real and generated images.

Zhu et al. [29] proposed CycleGAN, which makes the reconstructed image match well with the input image and ensures further model optimization by a cycle consistency constraint. To overcome texture differences, Kim and Byun [30] generated composite images with multiple different textures using style migration algorithms and used self-training to learn the target features. Choi et al. [31] proposed a target-guided and cycle-free data augmentation scheme, which generates images by adding semantic constraints to the generator and extracting style information from the target domain. The main idea of self-supervision methods is to reduce the source–target domain gaps by adding self-supervised learning tasks. The common auxiliary self-supervision tasks are image rotation prediction, puzzle prediction and position prediction [32,33].

In addition, there are some interesting UDA strategies, Sun et al. [34] proposed a discrepancy-based method, which minimizes domain shift by aligning the second-order statistics of source and target distributions. Yang et al. [35] proposed an adversarial agent that learns a dynamic curriculum for source samples, which improves the transferability of the domain by constantly updating the curriculum. Vu et al. [36] reduced the gap between the source and target domains by decreasing the entropy value of the target domain. Zou et al. [37] proposed a self-training based domain adaptive framework that unifies feature alignment and tasks.

Furthermore, UDA models have been widely applied for recognizing and interpreting remote sensing images to solve image discrepancy problems that destroy model adaptation [38–41]. In recent work, Iqbal and Ali [42] proposed weakly-supervised domain adaptation for built-up region segmentation, which achieved better adaptation and segmentation by adding weakly supervised tasks in the latent and output spaces. Similarly, Li et al. [43] proposed various weakly supervised constraints to reduce the disadvantageous influence of feature differences between the source and target domains, and multiple weakly-supervised constraints are composed of rotation consistency constraints, transfer invariant constraints, and pseudo-label constraints. Wittich and Rottensteiner [44] used adversarial training of appearance adaptation and classification networks instead of cycle consistency to constrain the model, which achieved good results in aerial image classification. Although these UDA methods have demonstrated excellent performance, single-target UDA models still have various challenges in multi-target tasks, and these limitations remain obstacles for semantic segmentation algorithms in practical applications.

### 2.1.2. Multi-Target Unsupervised Domain Adaptation

MTUDA transfers knowledge from a single source domain to multiple target domains. Most previous studies [45,46] focused on the classification task and less on MTUDA for segmentation. In general, MTUDA tasks can be divided into two cases: multiple implicit target domains and multiple explicit target domains. In the first category, the learner does not know which sub-target domain the relevant samples belong to during the entire domain adaptation process [47,48]. Due to domain gaps and categorical misalignments between multiple sub-target domains, the effectiveness of most existing DA models is greatly reduced. Chen et al. [47] proposed an adversarial meta-adaptive network with clusters generated from mixed target domain data as feedback to guide model learning.

In the second case, the sub-target domain identity of each sample is known during training but remains unknown during testing. To handle this issue, Nguyen-Meidine et al. [49] proposed the method that uses multiple teacher models to learn knowledge from different target domains, then extracts knowledge to a student model by knowledge distillation. Based on this concept, Isobe et al. [50] used data expanded by style transfer to train stronger teacher models, and a collaborative consistency learning strategy was designed to ensure full knowledge exchange among the target domains. Finally, a student model that worked well on multiple target domains was obtained by knowledge distillation with weight regularization. Gholami et al. [51] proposed a novel adversarial framework that achieved stronger connections between potential representations and target data by separating shared and private features and obtaining more robust feature representations.

Saporta et al. [20] proposed two MTUDA models for semantic segmentation and achieved great results. Lee et al. [52] proposed a multi-target domain transfer network to synthesize complex domain transferred images.

Overall, the current MTUDA algorithms for semantic segmentation are still in a considerably early stage and need further research.

### 2.2. Datasets

The study area is located in Yichang, China, where the fruit industry is an important economic pillar. As shown in Table 1, four remote sensing images include one unmanned aerial vehicle (UAV) image and three satellite images. They are acquired by different sensors at different times and locations, resulting in differences in illumination, resolution, and environmental conditions, which are factors contributing to the domain shift.

**Table 1.** The details of four remote sensing images.

| Index | ZG | CY | XT1 | XT2 |
|---|---|---|---|---|
| Source | UAV | Google Earth | Google Earth | DMC3 |
| Longitude and latitude | 110.5788E, 30.9798N | 111.3116E, 30.4194N | 111.5250E, 30.5663N | 111.5120E, 30.5543N |
| Spectral | RGB | RGB | RGB | RGB, NIR |
| Resolution | 0.2 m | 0.3 m | 0.6 m | 0.8 m |
| Image size | 13,000 × 10,078 | 12,000 × 12,000 | 10,000 × 12,000 | 10,000 × 10,000 |

To create multiple datasets, we process the four images in four steps. First, the near-infrared (NIR) band of XT2 is removed to ensure the same number of bands in all four images. Second, the complete image is divided into three parts, ensuring that there are no duplicates in the samples selected for the training set, validation set, and test set in each subsequent dataset. Third, we manually label three types of objects, including background pixels, orchard areas, and other vegetation. Finally, all samples are obtained by random cropping and data augmentation, which includes random rotation, flipping, and noise injection, and the sample size is set to 512 × 512.

It should be noted that the degree of our processing of the images is different due to the differences in image acquisition times. The training sets of Dataset ZG and Dataset CY are labeled and can be used as source domain; the training sets of Dataset XT1 and Dataset XT2 are unlabeled and can only be used as target domain. The validation and test sets in all datasets are labeled. The information of the four datasets is shown in Table 2. In addition, Figure 2 shows the orchard area samples derived from different datasets with significant visual differences.

**Table 2.** The four datasets for domain adaptation.

| Dataset | Total | Train Set | Validation Set | Test Set |
|---|---|---|---|---|
| Dataset ZG | 4598 | 2798 | 1200 | 600 |
| Dataset CY | 5000 | 3000 | 1200 | 800 |
| Dataset XT1 | 4162 | 2600 | 1062 | 500 |
| Dataset XT2 | 3500 | 2085 | 915 | 500 |

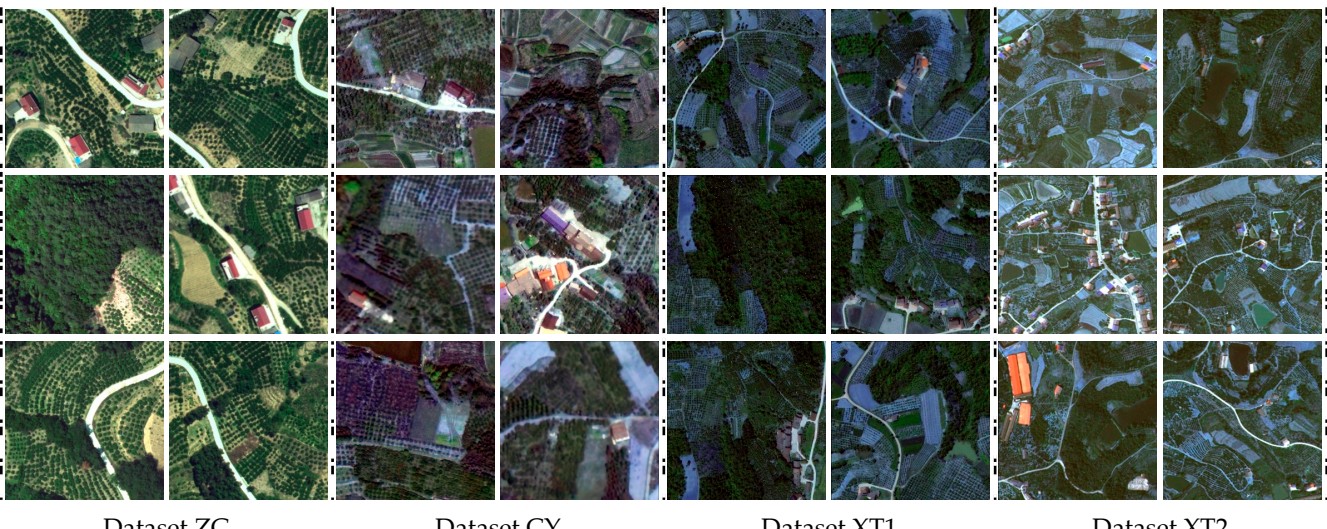

<div align="center">Dataset ZG      Dataset CY      Dataset XT1      Dataset XT2</div>

**Figure 2.** The figure displays the sample images from the four datasets.

### 2.3. Methods

#### 2.3.1. Preliminaries

Common UDA models have been designed for single-target domain settings. This paper considers a different UDA scenario, where the number K of target domains $K \geq 2$. We denote the data from the source domain as $X_S = \{(x_i, y_i)\}_{i=1}^{n_s}$, where $x_i$ and $y_i$ represent the source domain RGB image and the corresponding label, respectively, RGB is the color that represents the red, green and blue channels. The data from the target domains are denoted as $X_{T_k} = \left\{\left(z_j^k\right)\right\}_{j=1}^{n_{t_k}}$, $k = 1, \cdots, K$. $z_j^k$ represents the unlabeled image from the $k$-th target domain dataset, and $n_s$ and $n_{t_k}$ are the numbers of samples from different domains.

Two approaches are available for directly extending single-target UDA models to deal with data from multiple target domains. The first approach is to train the model separately for each target domain, but this approach is costly and difficult to scale. The second approach is to mix multiple target domains into a target domain to train a single-target UDA model; this approach ignores the inherent feature discrepancies among multiple target domains. To solve the interference between target domains, we propose a simple improvement scheme that constructs a model with multiple discriminators (Multi-D). Multi-D uses a discriminator for each source–target domain pair. The generator training process is affected by multiple adversarial losses and a segmentation loss, and the other aspects of Multi-D are similar to those of the fine-grained adversarial learning framework for domain adaptive (FADA) [22].

#### 2.3.2. MBUDA Network

From the perspective of different domains, the compositions of the features can be divided into shared (domain-invariant) features and private features. Previous work [20] focused on making the feature extractor learn the domain-invariant features while ignoring the importance of private features, which leads to negative migration and missing information. To solve these problems, we propose a novel MTUDA method. The multi-branch framework separates shared features and private features to prevent negative migration. The feature extractor learns more robust latent target domain data representations via multiple ancillary classifiers, providing sufficient classification information for the classifier. Figure 3 shows the MBUDA architecture, and the MBUDA training process is discussed below.

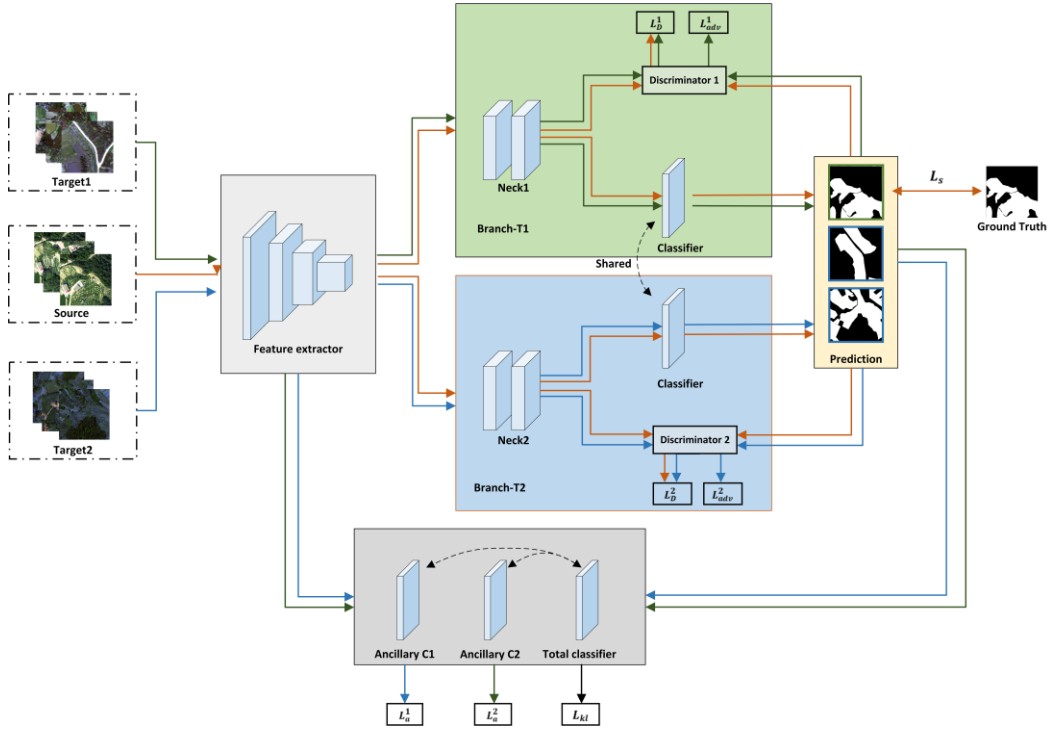

**Figure 3.** Overall architecture of the proposed MBUDA. The framework is illustrated with K = 2 as example but it also holds for other numbers of target domains. The feature extractor is optimized with segmentation loss $L_s^k$, ancillary segmentation loss $L_a^k$, and adversarial loss $L_{adv}^k$. The discriminator is optimized with $L_D^k$. The segmentation model is trained using different data, including data from the source domain (orange), target domain 1 (green), and target domain 2 (blue).

**Adaptation network training:** The source domain images are input into the feature extractor, and the $k$-th neck module generates the corresponding feature map $M_{s_k}$. Each neck module is a series of convolutional layers that help to separate shared features and private features and provide greater flexibility and control during information sharing. The feature extractor is the ResNet-101 [53] network that enables the gradient to flow freely. After that, the prediction can be obtained by inputting the feature map $M_{s_k}$ into classifier C. Because the source domain images have corresponding labels, the segmentation loss $L_s^k$ is defined by:

$$L_s^k = -\sum_{i \in n_s} \sum_{H,W} \sum_C y_i^{(H,W,C)} \log Q_{ik}^{(H,W,C)} \tag{1}$$

where $Q_{ik}$ is the prediction for source sample $i$ predicted by the $k$-th branch. The $k$-th target domain images are input into the feature extractor, and the $k$-th neck module and the $k$-th discriminator generate the prediction $H_{jk}$. Notably, the different branches only handle their corresponding target domains. Next, we can calculate the adversarial loss $L_{adv}^k$ according to the following formula:

$$L_{adv}^k = -\sum_{j \in n_{t_k}} \sum_{H,W} \sum_C a_{jk}^{(H,W,C)} \log H_{jk}^{(H,W,C)} \tag{2}$$

where $H_{jk}$ is the prediction for image $j$ derived from the $k$-th target domain. $a_{jk}$ is the domain label generated by classifier $C$, it is the same as multi-channel soft labels [22], which can guide the discriminator to learning better. The multiple ancillary classifiers are

designed to obtain more robust latent representations of target domain data. The ancillary segmentation loss for the $k$-th target domain is defined by:

$$L_a^k = - \sum_{j \in n_{t_k}} \sum_{H,W} \sum_C a_{jk}^{(H,W,C)} \log P_{jk}^{(H,W,C)} \tag{3}$$

where $P_{jk}$ is the binary segmentation output of image $j$ derived from the $k$-th target domain, which is generated by the $k$-th ancillary classifier. The overall cost function for the generator network is defined as follows:

$$L_G = \sum_{k=1,\cdots,K} \left( L_s^k + \lambda_{adv}^k L_{adv}^k + \gamma_a^k L_a^k \right) \tag{4}$$

where $\lambda_{adv}^k$ and $\gamma_a^k$ are the weight factors used to control the impacts of different losses. Since the information in the target domains is inaccessible during testing, we introduce a total classifier $C_{all}$ that learns all the knowledge possessed by other classifiers. Knowledge transfer is achieved via the Kullback–Leibler loss [54], which is defined by:

$$L_l = \sum_{k=1,\cdots,K} \sum_{j \in n_{t_k}} \sum_{H,W} \sum_C P_{jk}^{(H,W,C)} * \left( \log P_{jk}^{(H,W,C)} - \log I_{jk}^{(H,W,C)} \right) \tag{5}$$

where $I_{jk}$ is the binary segmentation output for image $j$ derived from the $k$-th target domain, which is generated by the total classifier $C_{all}$. Finally, the optimization function is defined as follows:

$$\min_G \max_D L(X_S, X_{T_k}) \tag{6}$$

**Discriminator training:** The discriminators are used to align features in different domains. Each discriminator only acts on its corresponding source–target domain pair. The discriminator loss is defined by:

$$L_D^k = - \sum_{i \in n_s} \sum_{H,W} \sum_C a_i^{(H,W,C)} \log H_{ik}^{(H,W,C)} - \sum_{j \in n_{t_k}} \sum_{H,W} \sum_C a_{jk}^{(H,W,C)} \log H_{jk}^{(H,W,C)} \tag{7}$$

where $H_{ik}$ is the prediction produced for source domain image $i$ in the $k$-th discriminator. $a_i$ is the relevant domain label. The total loss of multiple discriminators is defined as follows:

$$L_D = \sum_{k=1,\cdots,K} L_D^k \tag{8}$$

This work considers a variety of problems encountered by the common MTUDA model and further improves it.

### 2.3.3. Adaptation Enhanced Learning Strategy

In the domain adaptation task, it is necessary to consider the different domain gaps that are present. Pan et al. [25] reduced the intra-domain gaps by a self-supervised approach, but this method was designed based on a single-target UDA task. To reduce the target–target domain gaps, this paper designs an adaptation enhanced learning strategy to obtain a better adaptation effect. As illustrated in Figure 4, this strategy is divided into four steps. First, the MBUDA method is trained through the source domain and multiple target domains, the multiple discriminators in MBUDA directly reduce the source–target domain gaps and indirectly reduce the target–target domain gaps. In the second step, we use the trained MBUDA model and the entropy-based ranking function to obtain the prediction results and confidence scores of the target domain training set images, after that, we divide the prediction results of each target domain into a high-confidence part and a low-confidence

part according to the confidence scores. The confidence of image j from the *k*-th target domain is defined as follows:

$$E_{jk} = -\sum_{H,W} \sum_{C} P_{jk}^{(H,W,C)} \log P_{jk}^{(H,W,C)} \tag{9}$$

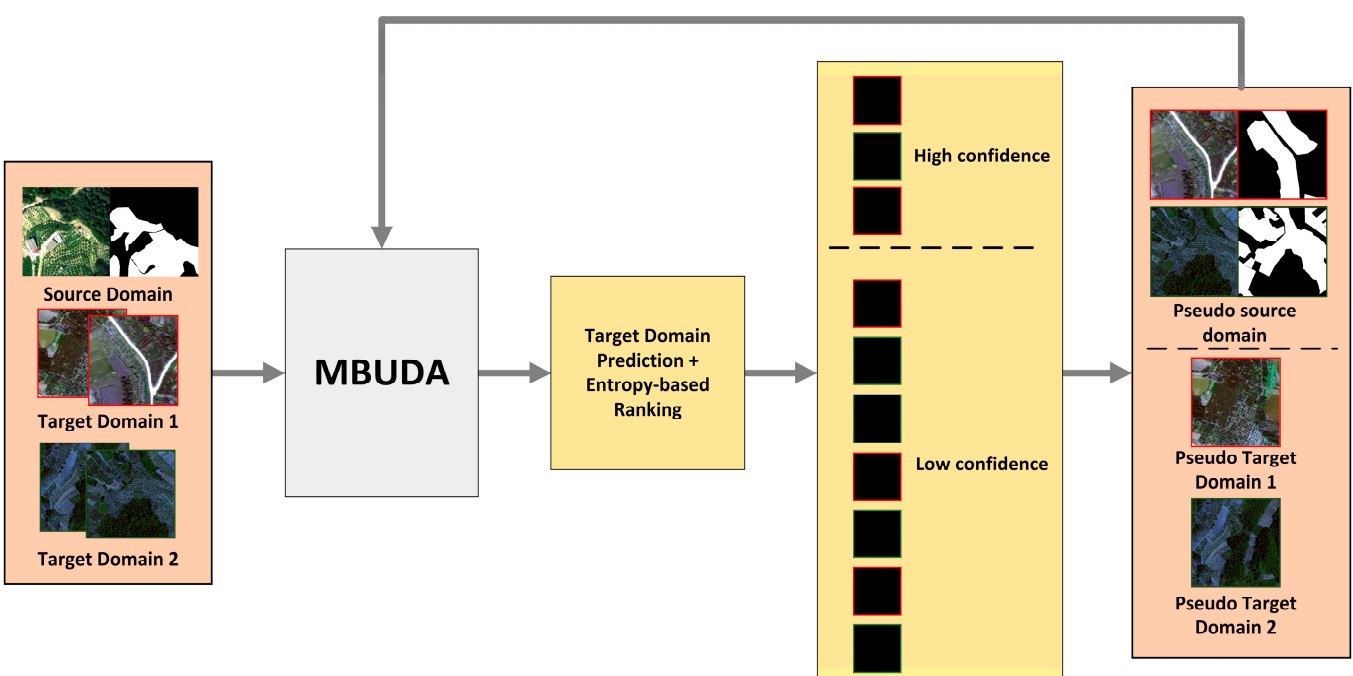

**Figure 4.** The overall framework of the adaptation enhanced learning strategy.

The data with high confidence can directly supervise the model to learn the potential features contained in multiple target domains. The data with low confidence have more noise and cannot be used directly. The third step is to mix all the data with high confidence as the pseudo-source domain and use the low-confidence part of the *k*-th target domain as the *k*-th pseudo-target domain without labels. The last step is to obtain the final model by retraining MBUDA with the pseudo-source domain and multiple pseudo-target domains. Notably, this paper considers three ways to use the obtained pseudo labels, and the details are described in the discussion.

## 3. Results

In this section, we first describe the detail of the experimental setup. Then, we introduce the evaluation metrics. Finally, we verify the effectiveness of the proposed models through a series of experiments.

### 3.1. Implementation Details

We use PyTorch deep learning framework to implement our method with a NVIDIA 3090 GPU having 24 GB of memory. We use the stochastic gradient descent (SGD) optimizer to train the generator; the momentum is 0.9, and the weight decay is $10^{-4}$. For the discriminator, the fine-grained discriminator in FADA is used as the discriminator in the Multi-D and MBUDA methods. The classifier is constructed via the ASPP [55] with the dilation rate set to [6,12,18,24]. The ratio of high-confidence and low-confidence parts is 3:7. The batch size is set to 12. The learning rate is initially set to $2.5 \times 10^{-4}$ and is adjusted according to a polynomial decay to the power of 0.9.

*3.2. Evaluation Metrics*

The evaluation metric is intersection over union (IoU) [56]. We have $IoU = \frac{TP}{TP+FP+FN}$, where TP, FN, FP are the numbers of true positive, false negative and false positive pixels, respectively. These calculations are based on the confusion matrix, and we will have a more intuitive understanding of the model effect through these indicators. To understand the segmentation results of the orchard area in the model directly, the results in the later experiments are based on the IoU of the orchard area, instead of the mean intersection over union over all classes.

*3.3. Experimental Results*

In the experiment, we consider and vary two factors: the type of domain shift and the number of target domains. Therefore, different domain combinations are selected in the four datasets to complete the transfer tasks. "Single-T Baselines" and "MT Baselines", referring to FADA [22], are the approaches, which directly extend single-target UDA models to deal with multiple target domain data. Multi-target knowledge transfer (MTKT) [20] is a MTUDA model. Enhanced multibranch unsupervised domain adaptation network (EMBUDA) is MBUDA with the addition of an adaptation enhanced learning strategy. Notably, the transfer task results are analyzed based on their average values.

3.3.1. Two Target Domains

According to the available data, six transfer tasks with different domain gaps are selected to prove the validity of the proposed model. There are six transfer tasks: (1) Dataset ZG to Dataset CY and Dataset XT1 (ZG → CY + XT1); (2) Dataset ZG to Dataset CY and Dataset XT2 (ZG → CY + XT2); (3) Dataset ZG to Dataset XT1 and Dataset XT2 (ZG → XT1 + XT2); (4) Dataset CY to Dataset ZG and Dataset XT1 (CY → ZG + XT1); (5) Dataset CY to Dataset ZG and Dataset XT2 (CY → ZG + XT2); (6) Dataset CY to Dataset XT1 and Dataset XT2 (CY → XT1 + XT2). Table 3 reports the results of three transfer tasks when Dataset ZG is the source domain. The segmentation model trained cannot obtain the expected results because of the domain gap, and the segmentation model without domain adaptation only achieves IoU of 42.17%, 23.90%, and 15.36% on Dataset CY, Dataset XT1, and Dataset XT2, respectively. The "Single-T Baseline" methods achieve IoU of 60.66%, 68.36%, and 63.66% on Dataset CY, Dataset XT1, and Dataset XT2, respectively, but multiple models are required for each domain, and the cost of training a model increases substantially when the number of target domains is too large. In comparison, "MT Baselines" directly combine the multiple target data as one domain but suffer considerable performance drops of 2.31%, 2.21%, and 1.84% on the three transfer tasks because of the domain shift between different target domains. The proposed methods show excellent performance compared to that of other MTUDA methods and the baselines. MBUDA achieves significant IoU gains of 6.61%, 8.92%, 6.72%, and 4.62% over the "Single-T Baselines", "MT Baselines", "MTKT", and "Multi-D" models, respectively, when adapting from Dataset ZG to Dataset CY, and Dataset XT1. Similarly, the proposed EMBUDA approach outperforms the "Single-T Baselines", "MT Baselines", "Multi-D", "MTKT", and MBUDA models by 7.88%, 10.19%, 5.89%, 7.99%, and 1.27% in terms of the IoU metric, respectively. On the other tasks, the proposed model also achieves good results.

Figures 5 and 6 show that the proposed models achieve better refinement results with better separation boundaries when adapting from Dataset ZG to Dataset CY, and Dataset XT1. More segmentation results can be found in Appendix A. Other MTUDA methods identify other vegetation as the orchard area because the features of both are similar and easily confused, which makes the adaptation process more difficult and causes the classifiers to need more information to make determinations. MBUDA considers the private features of different target domains and achieves a better adaptation effect.

**Table 3.** The IoU of orchard area segmentation when Dataset ZG is set as the source domain.

| Method | ZG → CY + XT1 | | | ZG → CY + XT2 | | | ZG → XT1 + XT2 | | |
|---|---|---|---|---|---|---|---|---|---|
| | CY | XT1 | Average | CY | XT2 | Average | XT1 | XT2 | Average |
| Without adaptation | 42.17 | 23.90 | 33.04 | 42.17 | 15.36 | 28.77 | 23.90 | 15.36 | 19.63 |
| Single-T Baselines | 60.66 | 68.36 | 64.51 | 60.66 | 63.66 | 62.16 | 68.36 | 63.66 | 66.01 |
| MT Baselines | 58.52 | 65.88 | 62.20 | 57.20 | 62.69 | 59.95 | 66.32 | 62.02 | 64.17 |
| MTKT | 58.05 | 70.75 | 64.40 | 58.14 | 58.07 | 58.11 | 67.77 | 58.78 | 63.28 |
| Multi-D | 62.99 | 70.01 | 66.50 | 59.18 | 64.43 | 61.81 | 68.06 | 62.61 | 65.34 |
| MBUDA | 66.59 | 75.65 | 71.12 | 66.91 | 70.01 | 68.46 | 74.76 | 71.91 | 73.34 |
| EMBUDA | **68.36** | **76.42** | **72.39** | **67.18** | **71.85** | **69.52** | **76.38** | **72.61** | **74.50** |

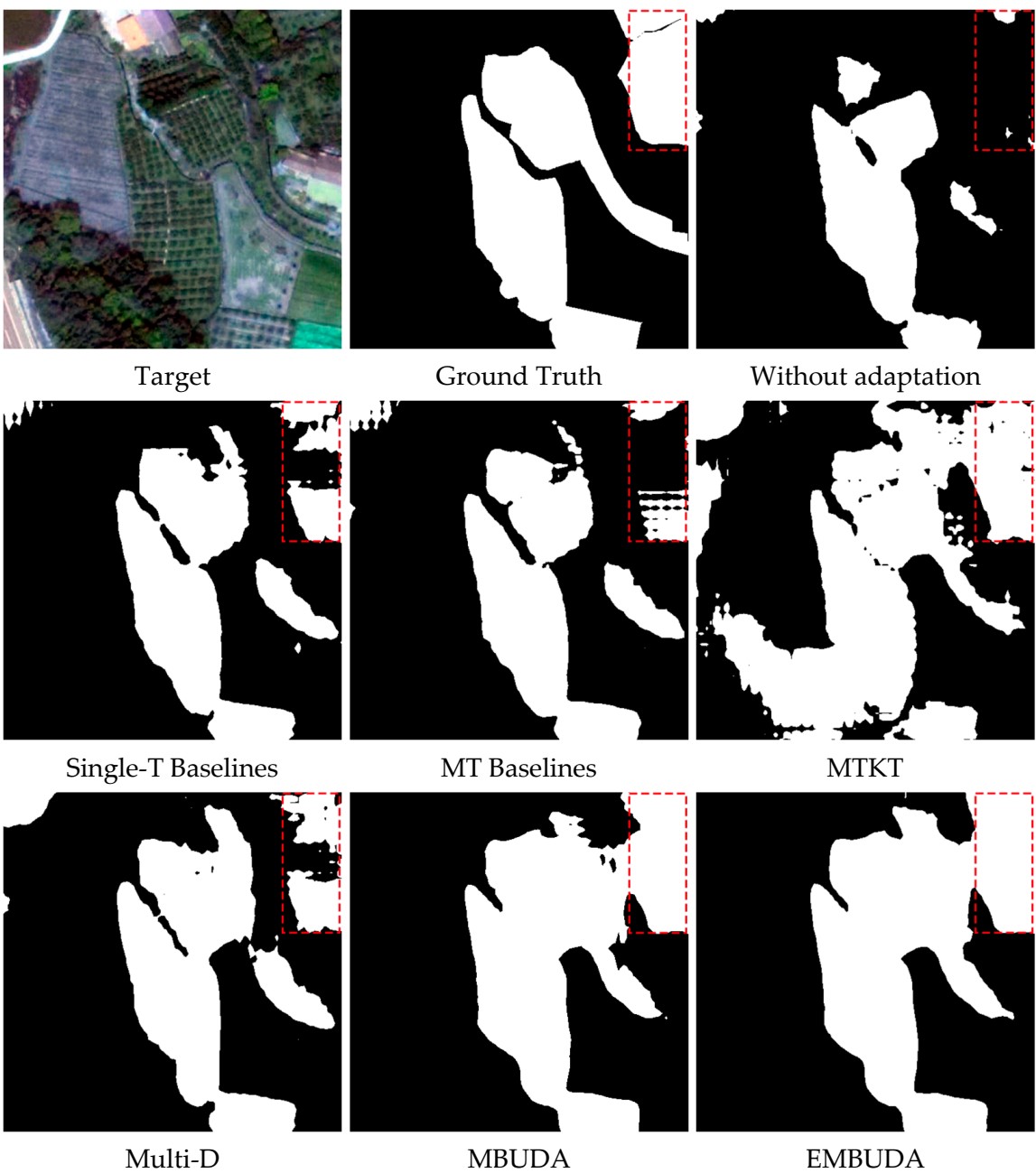

**Figure 5.** Outputs of orchard area segmentation in Dataset CY when adapting from Dataset ZG to Dataset CY and Dataset XT1.

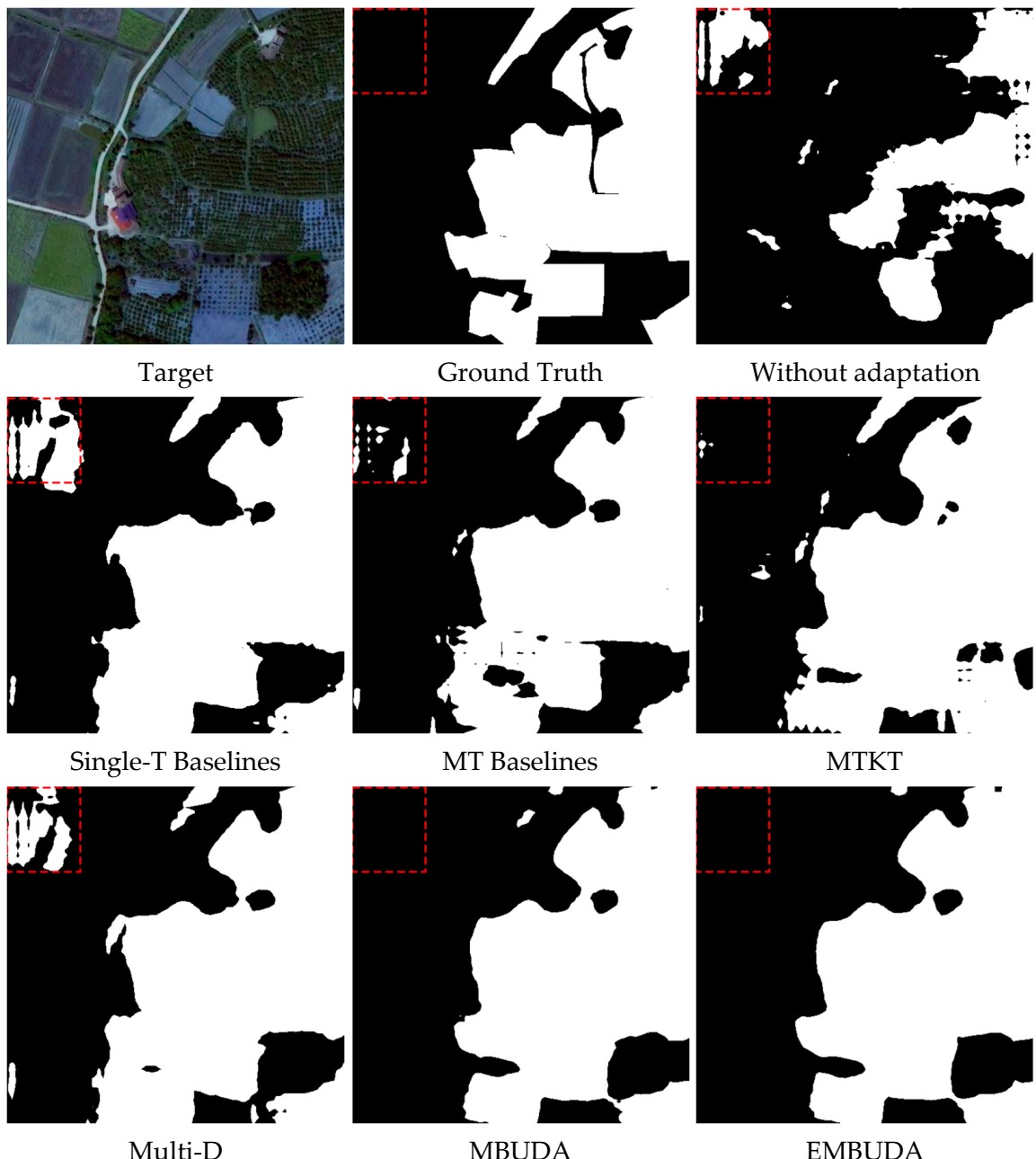

**Figure 6.** Outputs of orchard area segmentation in Dataset XT1 when adapting from Dataset ZG to Dataset CY and Dataset XT1.

Table 4 shows the results of orchard area segmentation when Dataset CY is the source domain. Interestingly, Dataset ZG achieves an IoU of 64.12% without domain adaptation. We conjecture that Image CY covers a wide area and there is diversity among the samples, which makes the domain gap on the model trained with Dataset ZG less significant. In addition, the proposed methods achieve state-of-the-art performance over the baselines and other existing methods. When adapting from Dataset CY to Dataset XT1, and Dataset XT2, MBUDA achieves 3.93%, 6.09%, 6.29%, and 3.49% IoU gains over the "Single-T Baselines", "MT Baselines", "MTKT", and "Multi-D", respectively. Similarly, the proposed EMBUDA outperforms the "Single-T Baselines", "MT Baselines", "MTKT", "Multi-D", and MBUDA by 4.99%, 7.15%, 4.55%, 7.35%, and 1.06% in terms of the IoU metric, respectively. The visualization results of transfer tasks can be seen in Appendix A.

**Table 4.** The IoU of orchard area segmentation when Dataset CY is set as the source domain.

| Method | CY → ZG + XT1 | | | CY → ZG + XT2 | | | CY → XT1 + XT2 | | |
|--------|------|------|---------|------|------|---------|------|------|---------|
| | ZG | XT1 | Average | ZG | XT2 | Average | XT1 | XT2 | Average |
| Without adaptation | 64.12 | 58.77 | 61.45 | 64.12 | 29.31 | 46.72 | 58.77 | 29.32 | 44.04 |
| Single-T Baselines | 66.30 | 67.02 | 66.66 | 66.30 | 68.61 | 67.46 | 67.02 | 68.61 | 67.82 |
| MT Baselines | 64.47 | 64.23 | 64.35 | 66.68 | 62.01 | 64.35 | 67.78 | 63.54 | 65.66 |
| MTKT | 65.75 | 68.16 | 66.96 | 64.99 | 59.06 | 62.03 | 71.05 | 59.86 | 65.46 |
| Multi-D | 64.25 | 66.82 | 65.54 | 64.74 | 67.06 | 65.90 | 68.10 | 68.42 | 68.26 |
| MBUDA | 75.78 | 72.52 | 74.15 | 74.39 | 71.98 | 73.19 | 73.21 | 70.29 | 71.75 |
| EMBUDA | **76.36** | **73.12** | **74.74** | **74.84** | **73.27** | **74.06** | **74.71** | **70.91** | **72.81** |

### 3.3.2. Three Target Domains

To further illustrate the validity of the MBUDA and EMBUDA approaches, we consider a more challenging setup (K = 3). The increase in the number of target domains means that more uncertainty is contained in the adaptation process; therefore, the process difficulty is also increased. Table 5 describes the results produced by the proposed methods and the other MTUDA methods. When Dataset ZG is used as the source domain, MBUDA achieves 43.69%, 6.6%, 9.42%, 8.69%, and 5.78% IoU gains over the "Without Adaptation", "Single-T Baselines", "MT Baselines", "MTKT", and "Multi-D" methods, respectively. The proposed EMBUDA approach outperforms the "Without Adaptation", "Single-T Baselines", "MT Baselines", "MTKT", "Multi-D", and MBUDA methods by 44.78%, 7.69%, 10.51%, 9.78%, 6.87%, and 1.09% in terms of the IoU metric, respectively. When Dataset ZG is used as the source domain, The proposed EMBUDA approach outperforms the "Without Adaptation", "Single-T Baselines", "MT Baselines", "MTKT", "Multi-D", and MBUDA methods by 25.00%, 8.42%, 11.35%, 10.55%, 9.14%, and 1.49% in terms of the IoU metric, respectively. As shown in Figures 7–9, the proposed methods achieve better segmentation results than the other methods. Common multi-target unsupervised domain adaptation methods easily confuse the class features in the orchard area segmentation task. MBUDA learns more robust latent representations of the target domain data and has a superior ability to discriminate between class features in the target domain.

**Table 5.** The IoU of orchard area segmentation when the number of target domains is set to 3.

| Method | ZG → CY + XT1 + XT2 | | | | CY → ZG + XT1 + XT2 | | | |
|--------|------|------|------|---------|------|------|------|---------|
| | CY | XT1 | XT2 | Average | ZG | XT1 | XT2 | Average |
| Without adaptation | 42.17 | 23.90 | 15.36 | 27.14 | 64.12 | 58.77 | 29.31 | 50.73 |
| Single-T Baselines | 60.66 | 68.36 | 63.66 | 64.23 | 66.30 | 67.02 | 68.61 | 67.31 |
| MT Baselines | 56.02 | 66.52 | 61.68 | 61.41 | 64.40 | 66.19 | 62.55 | 64.38 |
| MTKT | 58.33 | 68.89 | 59.20 | 62.14 | 64.18 | 69.61 | 61.74 | 65.18 |
| Multi-D | 62.79 | 69.89 | 62.48 | 65.05 | 64.88 | 68.36 | 66.53 | 66.59 |
| MBUDA | 64.97 | 74.83 | 72.69 | 70.83 | 77.30 | 73.51 | 71.91 | 74.24 |
| EMBUDA | **66.21** | **75.62** | **73.92** | **71.92** | **79.69** | **75.36** | **72.14** | **75.73** |

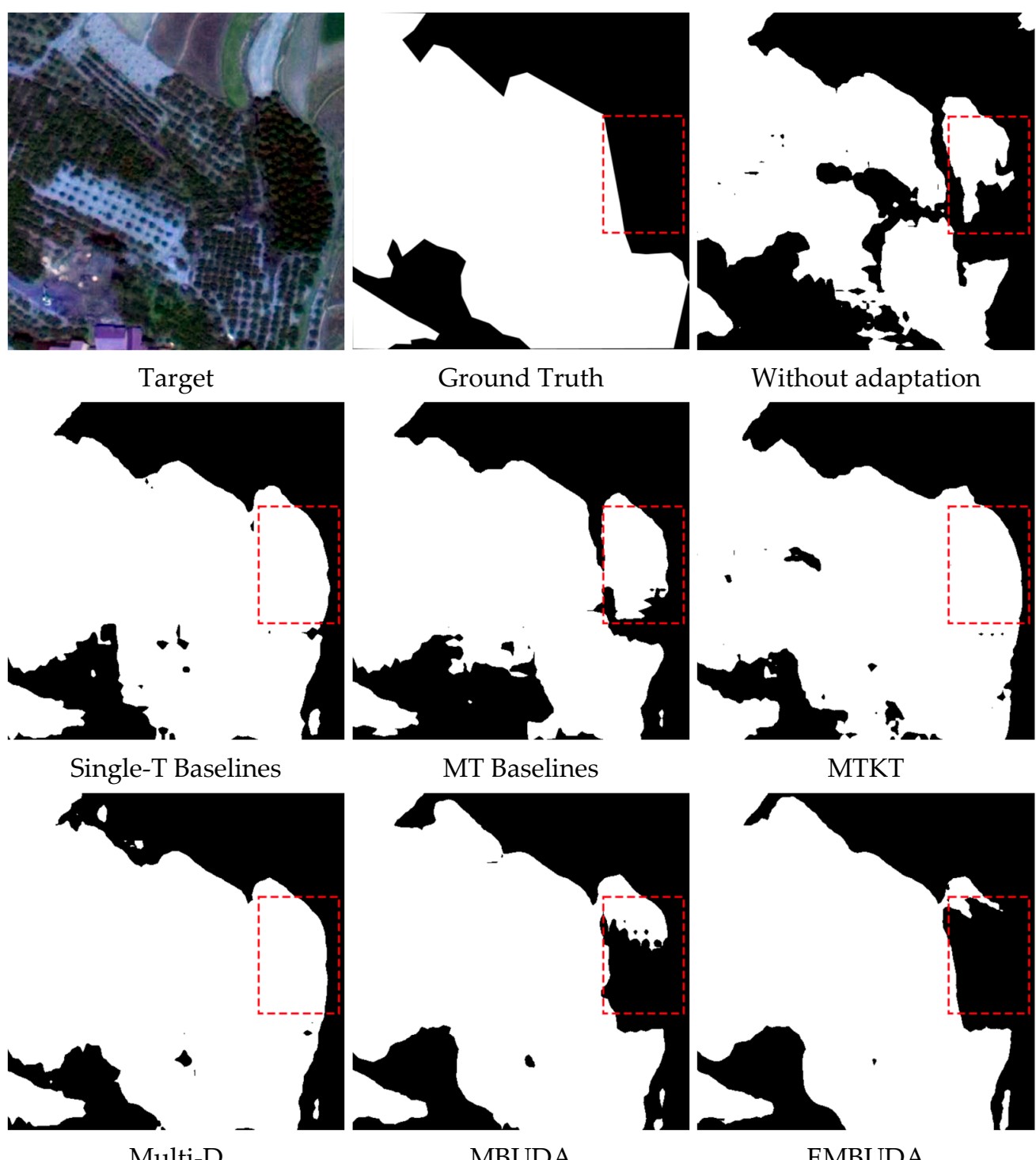

**Figure 7.** Outputs of orchard area segmentation in Dataset CY when adapting from Dataset ZG to Dataset CY, Dataset XT1, and Dataset XT2.

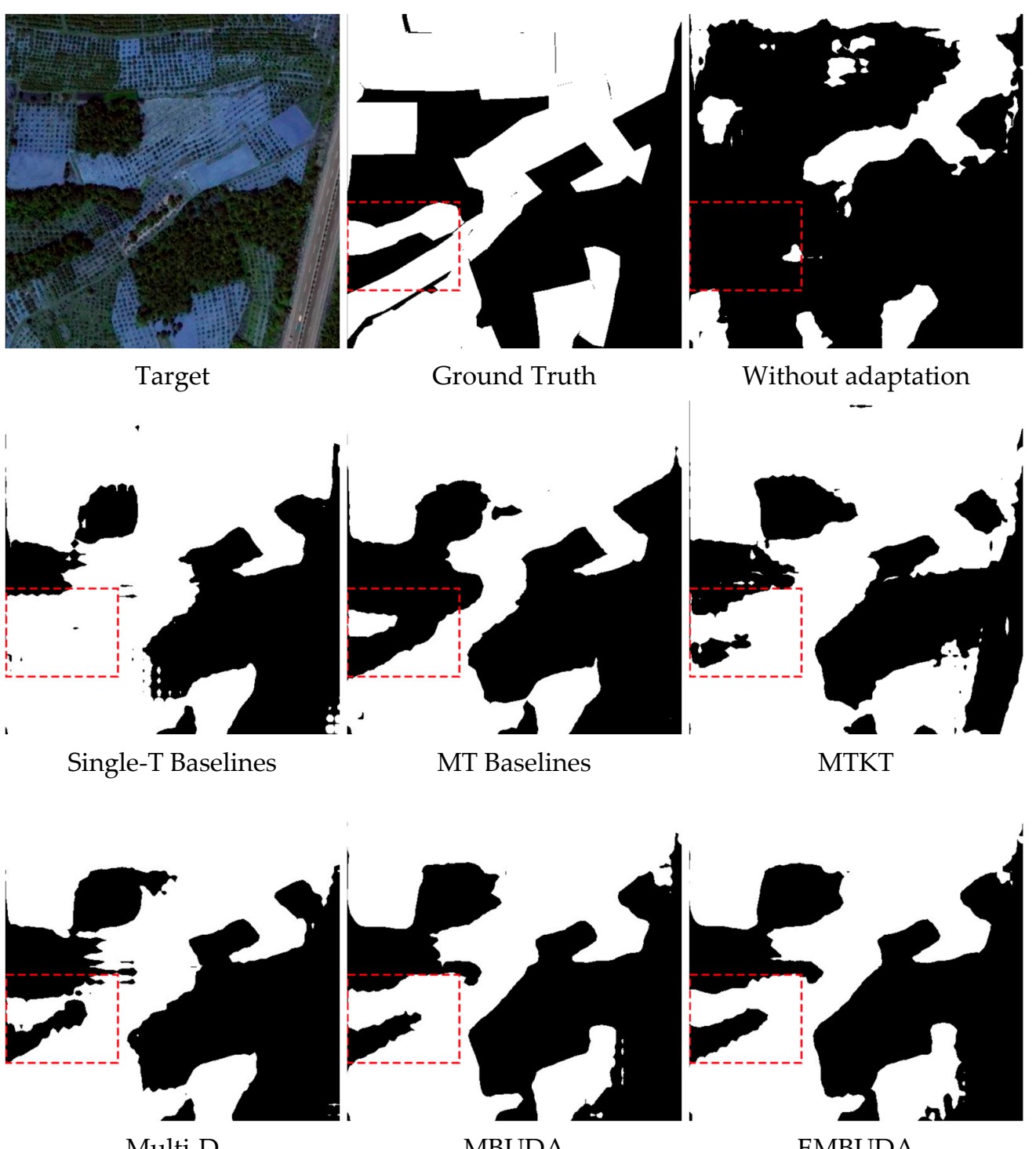

**Figure 8.** Outputs of orchard area segmentation in Dataset XT1 when adapting from Dataset ZG to Dataset CY, Dataset XT1, and Dataset XT2.

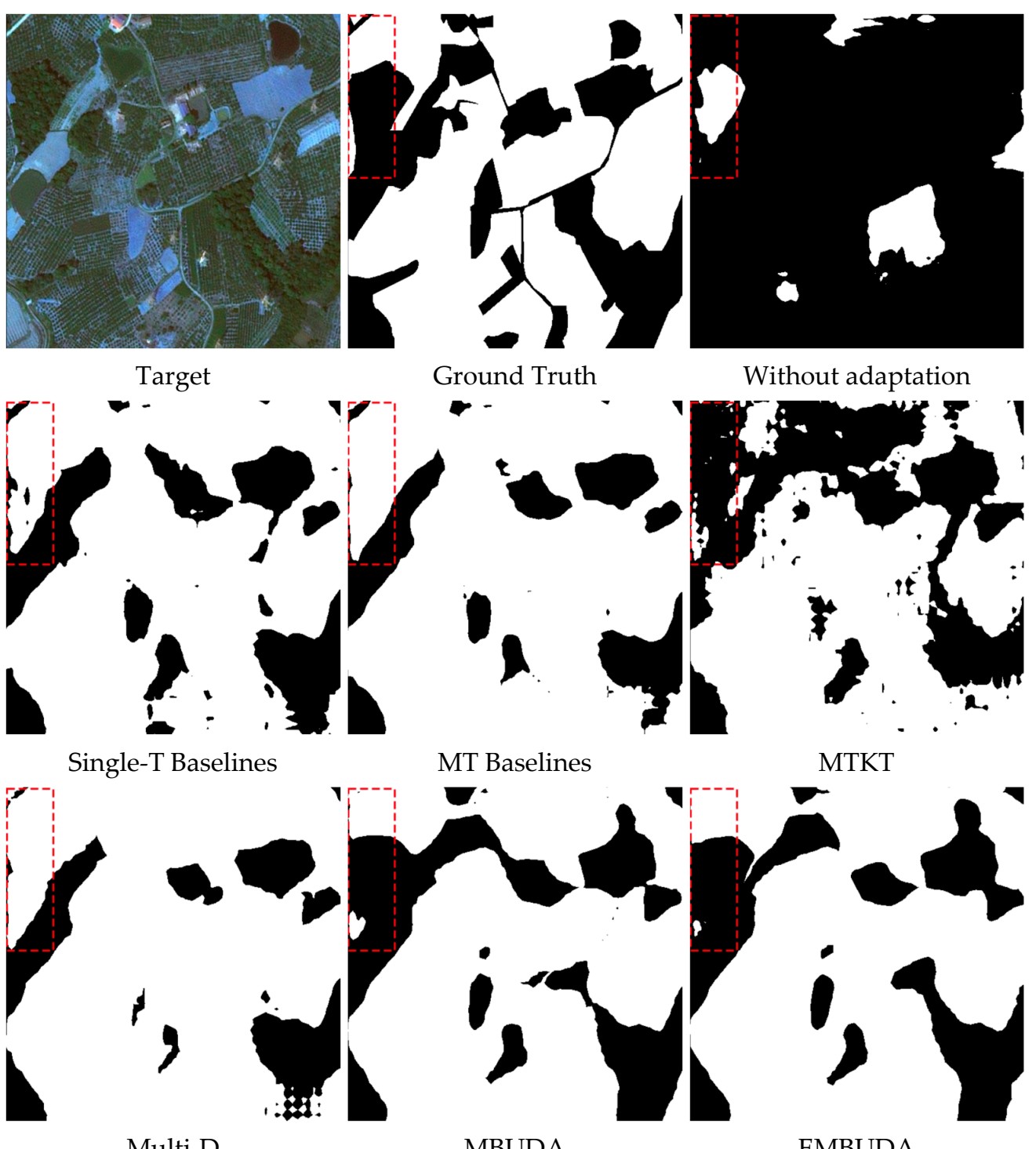

**Figure 9.** Outputs of orchard area segmentation in Dataset XT2 when adapting from Dataset ZG to Dataset CY, Dataset XT1, and Dataset XT2.

### 3.4. Additional Impact of Pseudo Labels

MBUDA handles different source–target domain pairs separately through multiple branches, but the model only indirectly reduces the target–target domain gaps through the source domain linkage during the training process.

To further reduce the target–target domain gaps, we consider three approaches to extend pseudo labels to the MBUDA approach. (1) The $k$-th target domain images are divided into two parts (high-confidence and low-confidence parts), and the two parts of

the kth target domain are aligned on the *k*-th branch. (2) The high-confidence images from the multiple target domains in approach 1 are combined into a pseudo-source domain, the low-confidence parts of different target domains are used as different-pseudo target domains, and the method is trained in the same way as MBUDA. (3) The confidence levels of all target domain images are ranked; this is the EMBUDA method. PL1, PL2, and PL3 in Table 6 correspond to the above three approaches. As shown in Table 6, all three approaches obtain performance improvements, and the third approach achieves the best result.

**Table 6.** Additional impact of pseudo labels. the results are based on the IoU of the orchard area.

| Method | ZG → CY + XT1 | | | ZG → CY + XT1 + XT2 | | | |
| --- | --- | --- | --- | --- | --- | --- | --- |
| | CY | XT1 | Average | CY | XT1 | XT2 | Average |
| MBUDA | 66.59 | 75.65 | 71.12 | 64.97 | 74.83 | 72.69 | 70.83 |
| MBUDA+PL1 | 67.20 | 76.33 | 71.77 | 66.01 | **75.77** | 73.29 | 71.69 |
| MBUDA+PL2 | 67.43 | **76.48** | 71.96 | 66.12 | 75.29 | 73.57 | 71.66 |
| MBUDA+PL3 | **68.36** | 76.42 | **72.39** | **66.21** | 75.62 | **73.92** | **71.92** |

## 4. Discussion

### 4.1. Comparison of Different Models

The proposed models achieve better results in the multi-group domain adaptive task, which prove the effectiveness of the proposed models. Specifically, the proposed methods have the following three advantages:

1. Traditional domain adaptive models reduce the domain gap with the goal of adapting from the source to a specific target domain. When multiple target domain data exist, as described in Section 2.3.1, there are two ways to directly extend single-target UDA models to work on multiple target domains, but the results of these methods are not satisfactory. As shown in Table 3, the "Single-T Baselines" approach is costly and difficult to scale, and "MT Baselines" ignores distribution shifts across different target domains. The proposed MBUDA handles source–target domain pairs separately by multiple branches, which enables the source domain to align multiple target domains simultaneously, and MBUDA simplifies the training process while ensuring the performance of the segmentation model;

2. In MTUDA task, segmentation models need to learn the full potential representation of the target domain in order to better predict images from different domains. The proposed MBUDA separates the feature learning and alignment processes, which prevents private features to interfere the alignment and ensures that the model learns both invariant features and private features. In Table 3, MBUDA achieves 4.62%, 6.65%, and 8.00% IoU gains over the "MT Baselines" on the three groups of tasks, respectively. We attribute the improved performance to the better feature representation learned by the model. As shown in Figure 6, the proposed methods have better discriminative ability for other classes and orchard areas;

3. Most unsupervised domain adaptive models do not consider the large distribution gap in the target domain itself during the process of aligning source–target domain features. In this paper, we design an adaptation enhanced learning strategy to use pseudo labels to directly reduce the target–target domain gaps. As shown in Table 3, EMBUDA achieves 1.27%, 1.06%, and 1.16% IoU gains over the MBUDA on the three groups of tasks, respectively. The improvement in model performance demonstrates the importance to further reduce the target–target domain gaps.

### 4.2. Impact of Training Data

In the different domain adaptation tasks, the performance of MTUDA models on the same test set varies, which is caused by different training data. As shown in Tables 3 and 4, there are significant differences in the performance of the models on the target domain when the target domain is the same and the source domain is different. For example, the

target domains are Dataset XT1 and Dataset XT2, "MT Baselines", "MTKT", "Multi-D", MBUDA, and EMBUDA have a change of 1.49%, 2.18%, 3.12%, 1.59%, and 1.69% in terms of the IoU metric, respectively. This phenomenon is caused by the domain gaps in different source and target domains, which leads to the feature alignment with different degrees of difficulty.

In the other case, the interference with the models come from the inherent discrepancy among target domains. As shown in the latter two sets of domain adaptation tasks in Table 3, there are differences in the performance of multiple MTUDA models on Dataset XT2, "MTKT", "Multi-D", and MBUDA have a change of 0.71%, 1.82%, and 3.45% in terms of the IoU metric, respectively. The last case we consider is that the domains in the training data are the same, but the roles are different. In this case, both the different effects of source–target domain feature alignment and the inherent differences between the target domains affect the training of the MTUDA models.

Although the training data significantly interferes with the model performance, our proposed model performs well in several tasks, which proves that our approach can adapt to the changes brought by the training data.

*4.3. Future Work*

In this paper, we have experimented on four datasets and the proposed model obtained better segmentation results. However, there are many difficulties that need to be explored in practical applications, for example, the case of more target domains and the case of cluttered target domains. In the future, we plan to build larger datasets containing more domain data and then experiment on larger datasets; on the other hand, the research scenario in this paper is the origin of known training samples. When the domain labels are unknown during training, the interference from the distribution gap between target domains is greater, which requires us to design special networks for the new scenarios.

## 5. Conclusions

In this study, we propose a novel MTUDA model for cross multidomain orchard area segmentation, which achieves good results on multiple target domains. The proposed multibranch framework is designed to separate shared features and private features, while preventing mutual interference between target domains. By adding multiple ancillary classifiers, the feature extractor learns the latent representation of the target domain data that is most conducive to classification. Furthermore, the proposed adaptation enhanced learning strategy further aligns the target–target domain features and enables the segmentation model to achieve better results in multi-target domain scenarios. Comprehensive experiments show that the general MTUDA schemes suffer from high costs and instability, while the proposed methods obtain better results on multiple-group transfer tasks than the single-target UDA baselines. Furthermore, although we conduct this study by focusing on orchard area segmentation, the proposed methods are not limited to specific types; thus, we have reason to expect that the proposed methods will also perform well in other semantic segmentation applications.

**Author Contributions:** Data curation, M.L.; Formal analysis, M.L.; Funding acquisition, D.R.; Investigation, H.S. and S.X.Y.; Methodology, M.L.; Supervision, H.S. and S.X.Y.; Writing—original draft, M.L.; Writing—review and editing, D.R. and H.S. All authors have read and agreed to the published version of the manuscript.

**Funding:** This work is supported by the National Key Research and Development Program of China (Grant No. 2016YFD0800902).

**Institutional Review Board Statement:** Not applicable.

**Informed Consent Statement:** Not applicable.

**Data Availability Statement:** The data are not publicly available because it is commercial data. The code and model are available at https://github.com/LM98mzhnq/MBUDA.git (accessed on 25 September 2022).

**Conflicts of Interest:** The authors declare no conflict of interest.

## Appendix A. Outputs of Orchard Area Segmentation

Figures A1–A9 show the results in different transfer tasks.

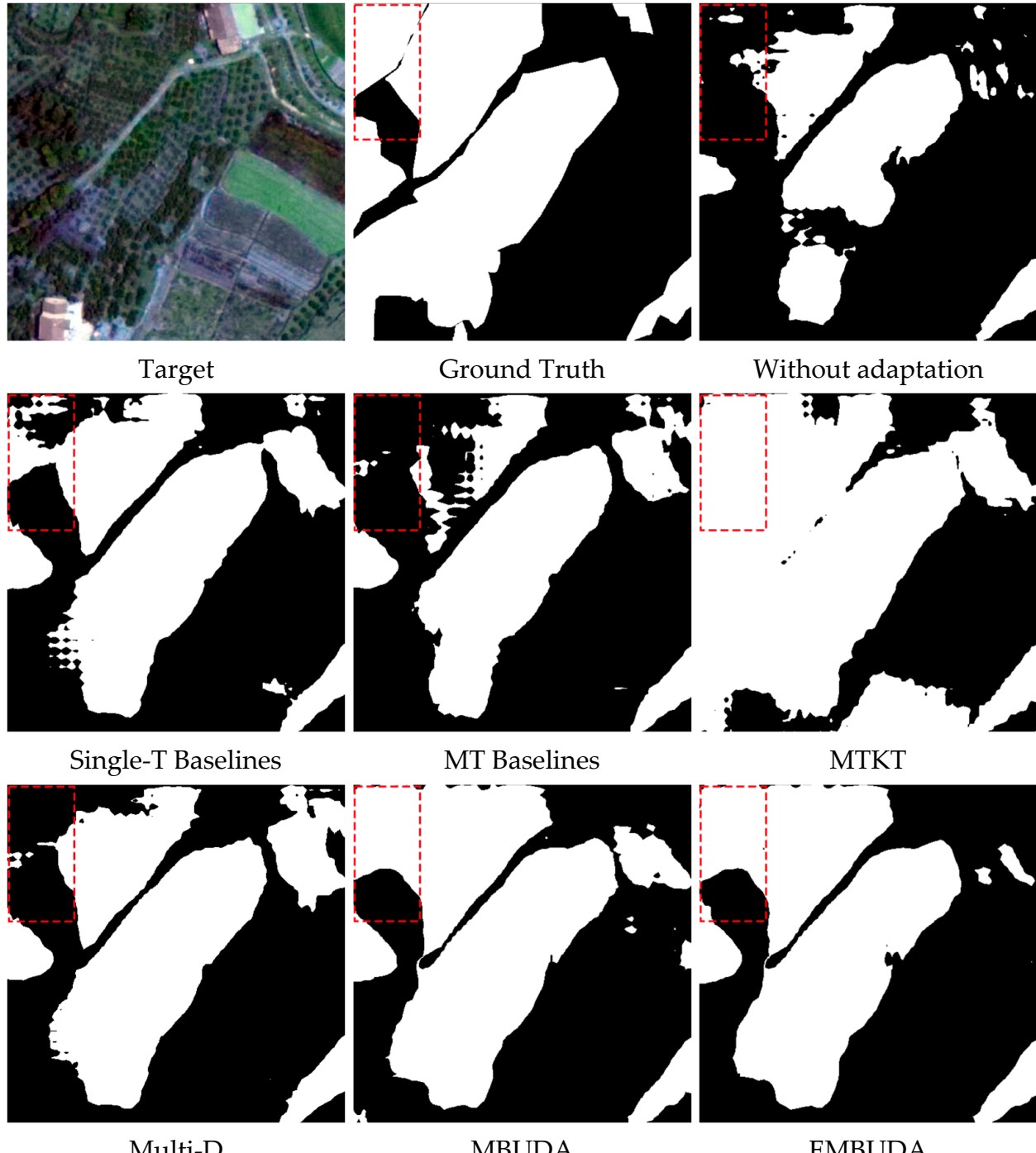

**Figure A1.** Outputs of orchard area segmentation in Dataset CY when adapting from Dataset ZG to Dataset CY and Dataset XT2.

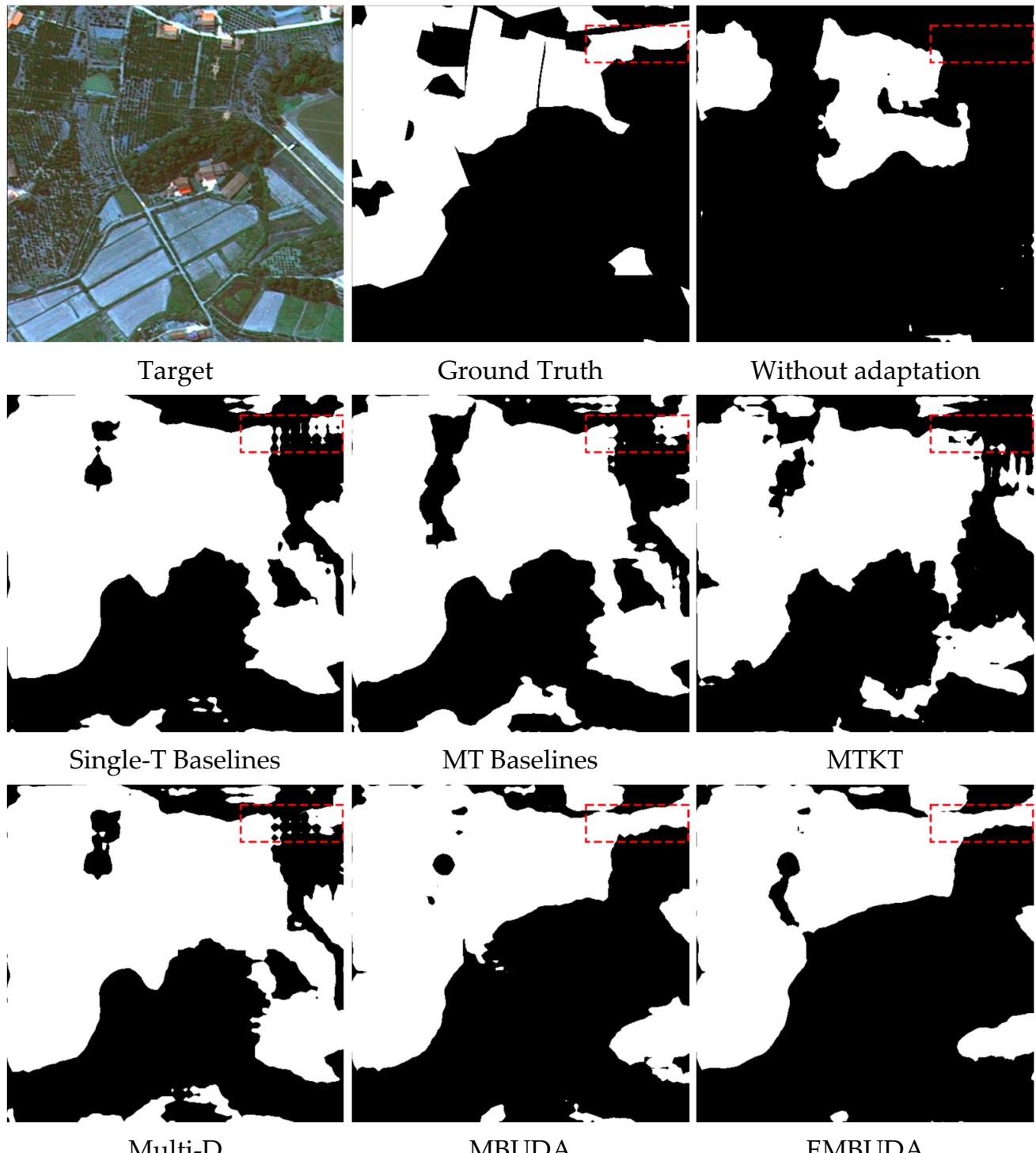

**Figure A2.** Outputs of orchard area segmentation in DatasetXT2 when adapting from Dataset ZG to Dataset CY and Dataset XT2.

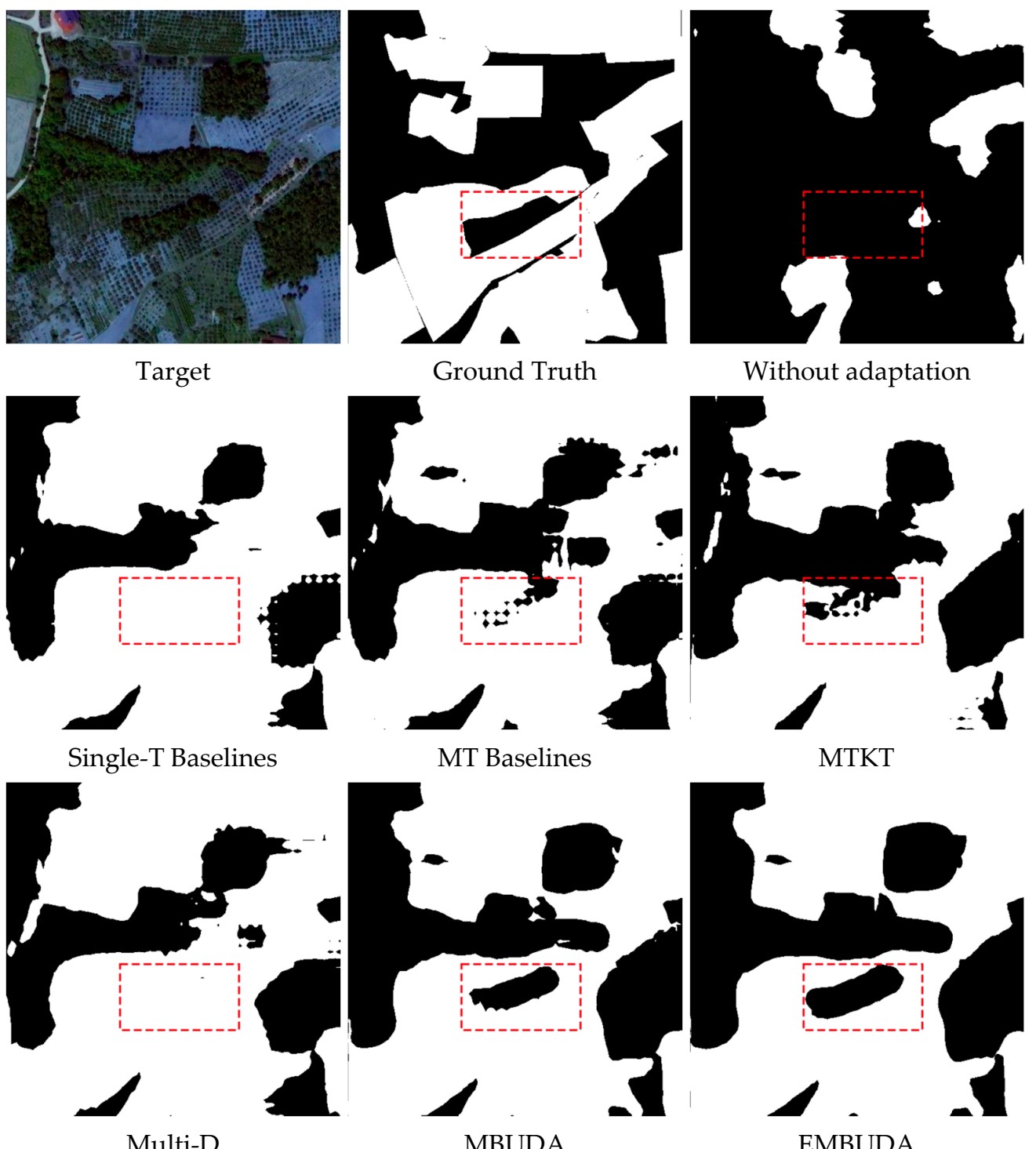

**Figure A3.** Outputs of orchard area segmentation in Dataset XT1 when adapting from Dataset ZG to Dataset XT1 and Dataset XT2.

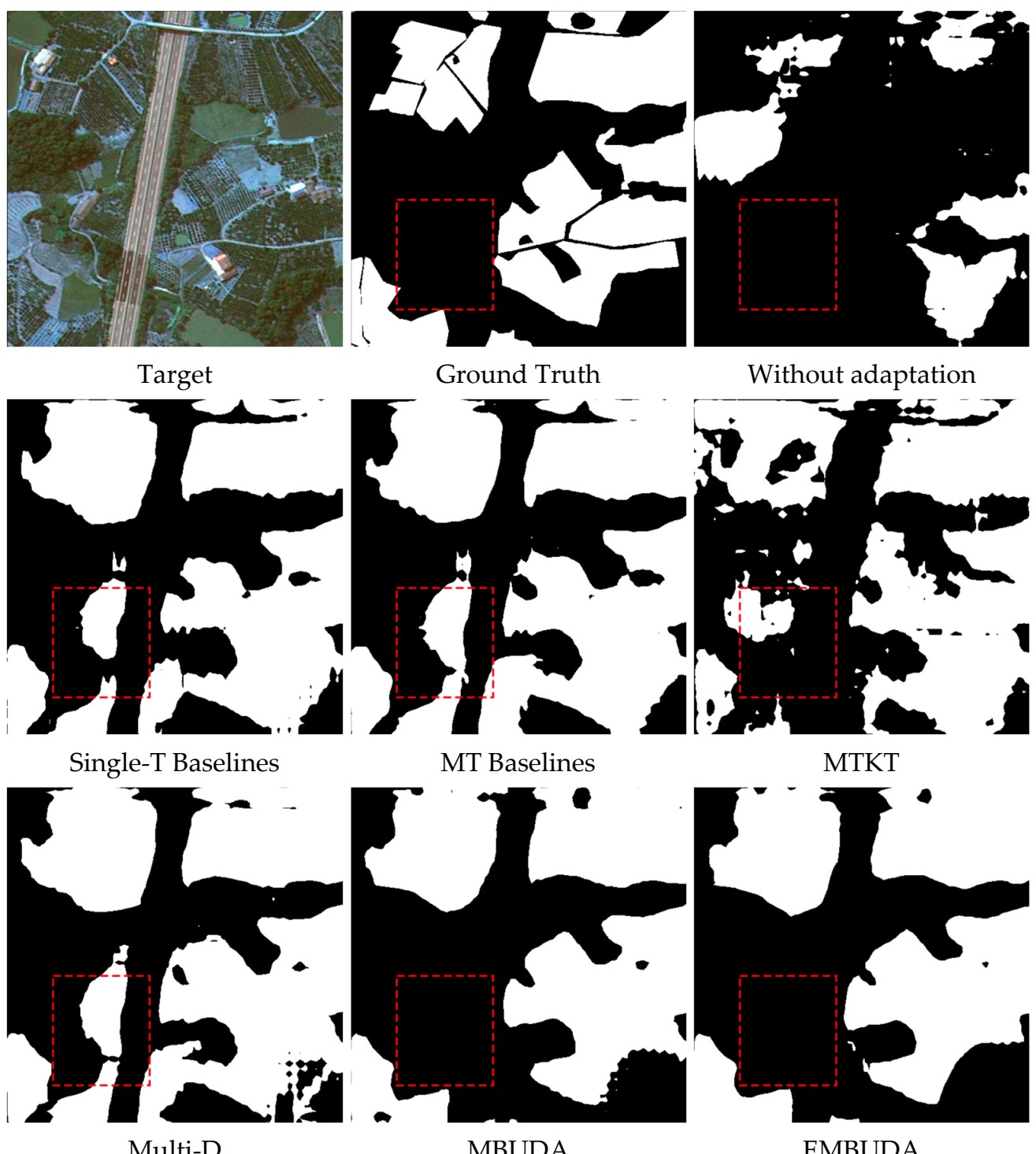

**Figure A4.** Outputs of orchard area segmentation in Dataset XT2 when adapting from Dataset ZG to Dataset XT1 and Dataset XT2.

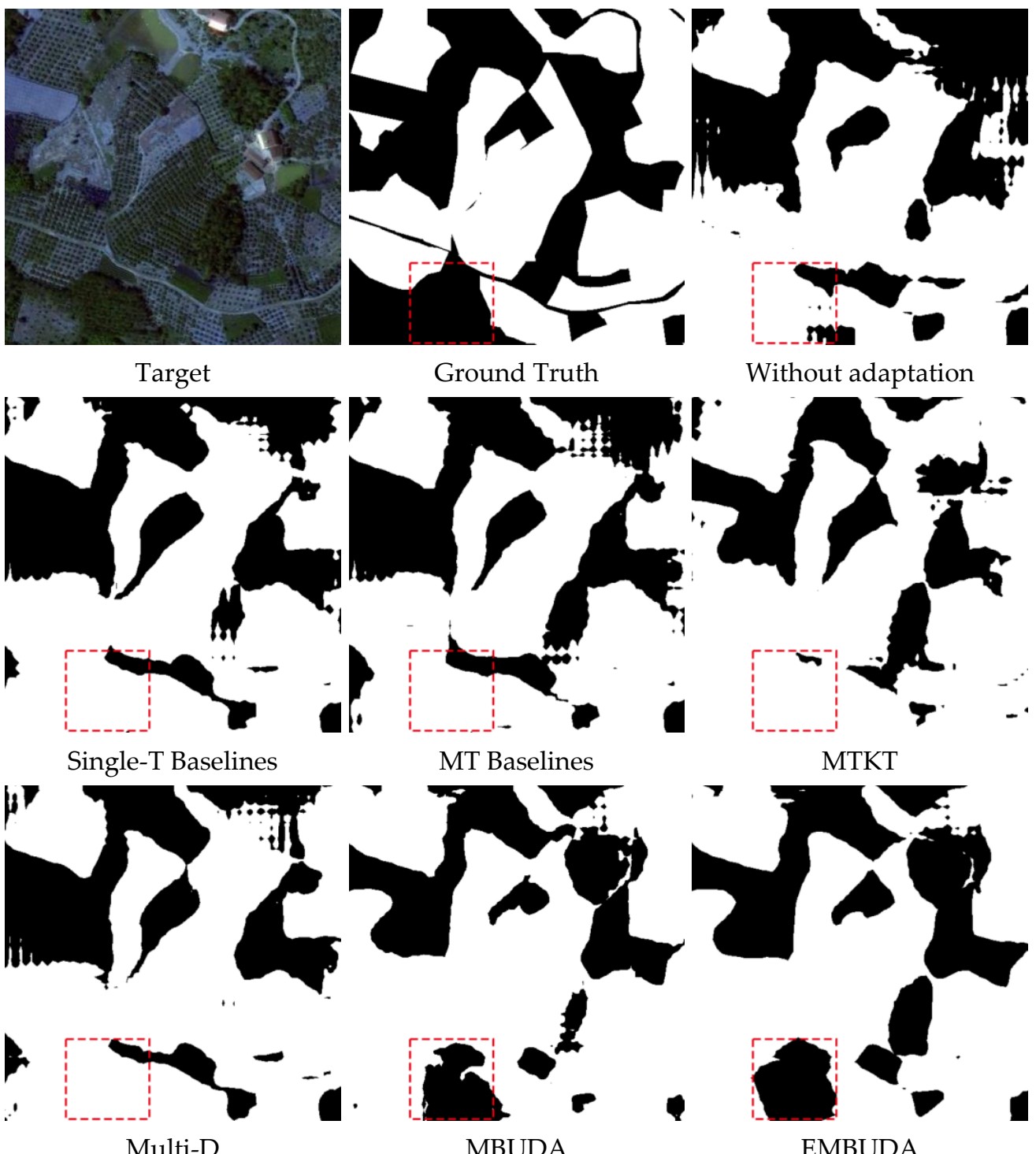

**Figure A5.** Outputs of orchard area segmentation in Dataset XT1 when adapting from Dataset CY to Dataset XT1 and Dataset XT2.

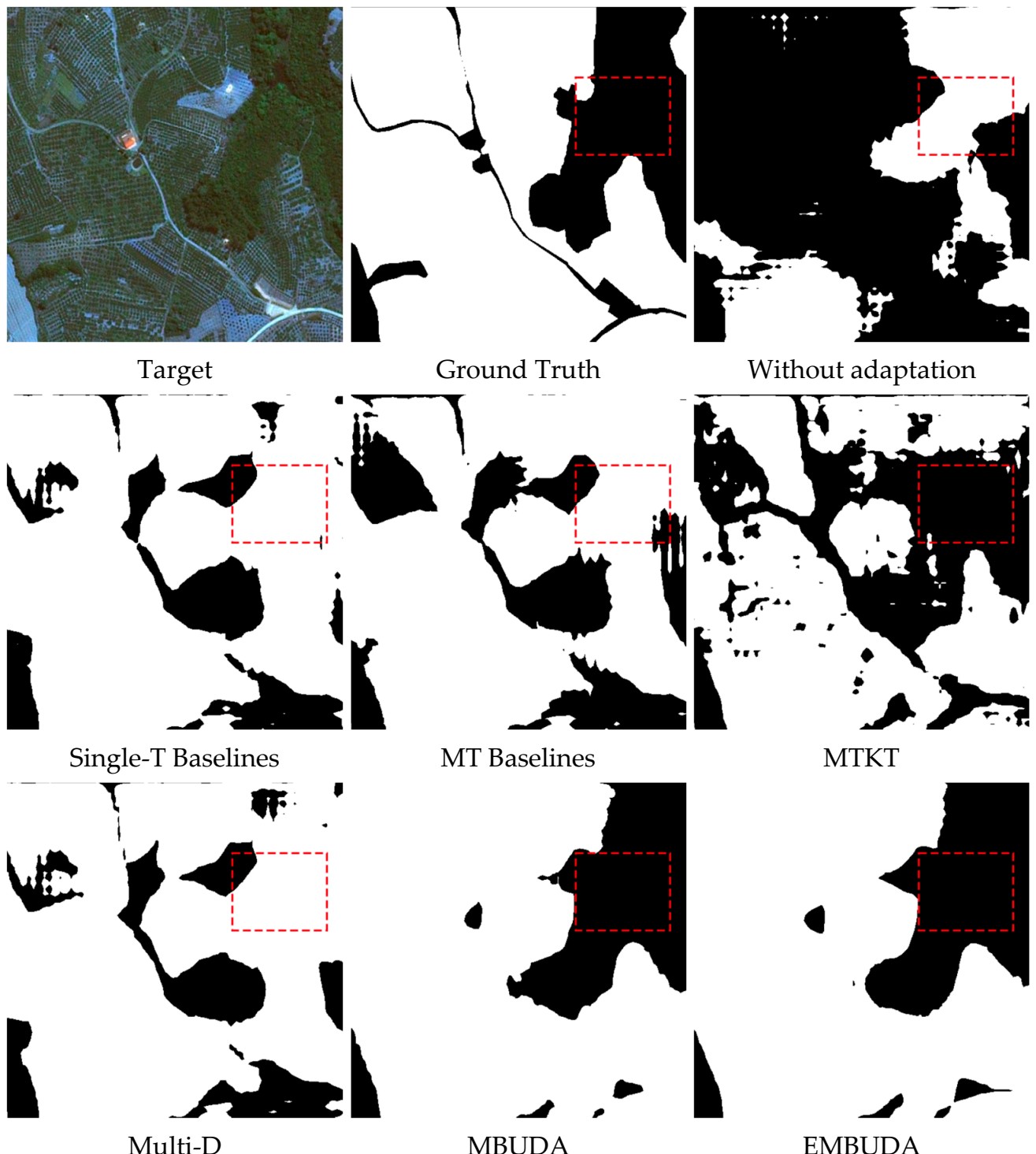

**Figure A6.** Outputs of orchard area segmentation in Dataset XT2 when adapting from Dataset CY to Dataset XT1 and Dataset XT2.

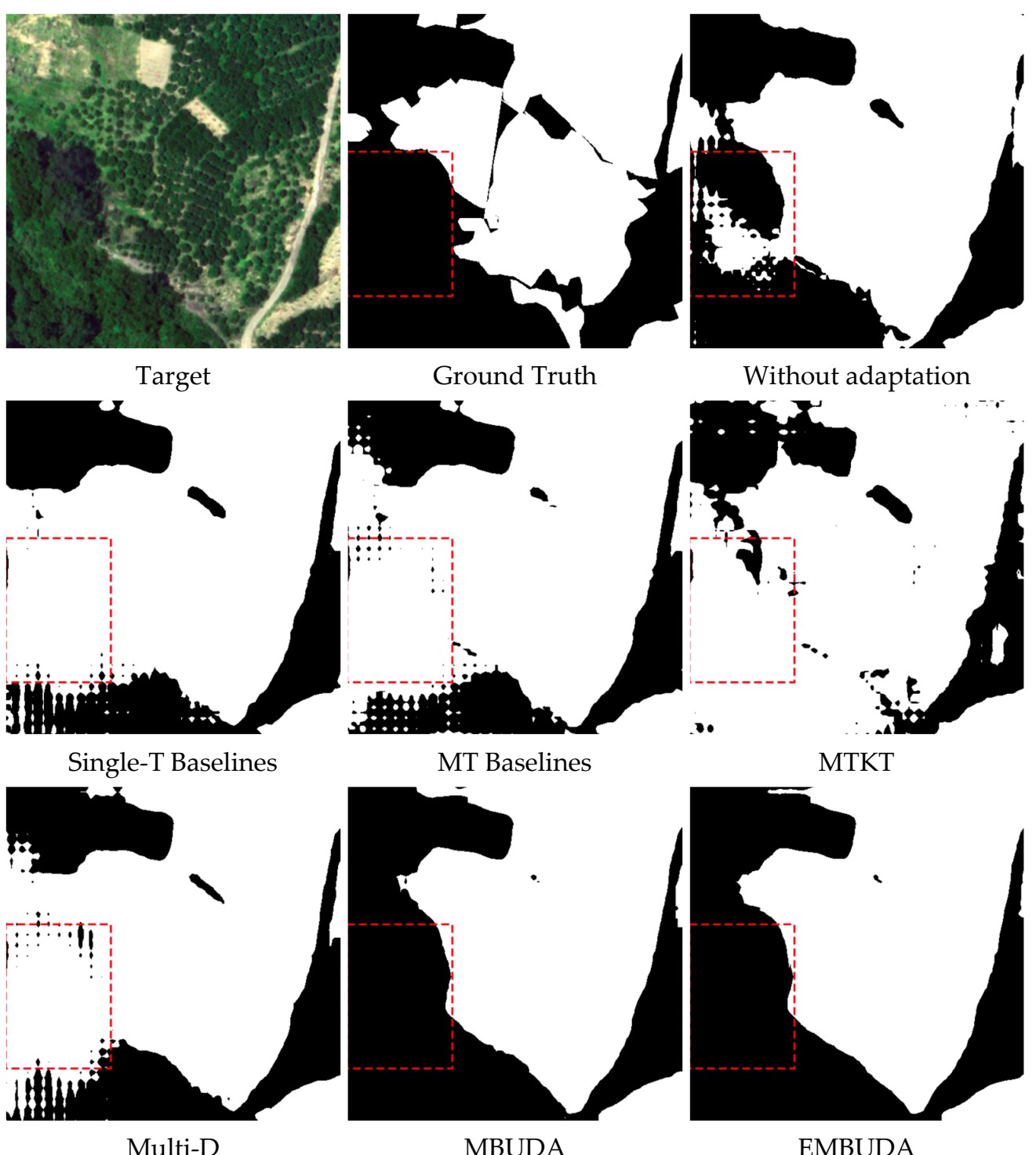

**Figure A7.** Outputs of orchard area segmentation in Dataset ZG when adapting from Dataset CY to Dataset ZG, Dataset XT1 and Dataset XT2.

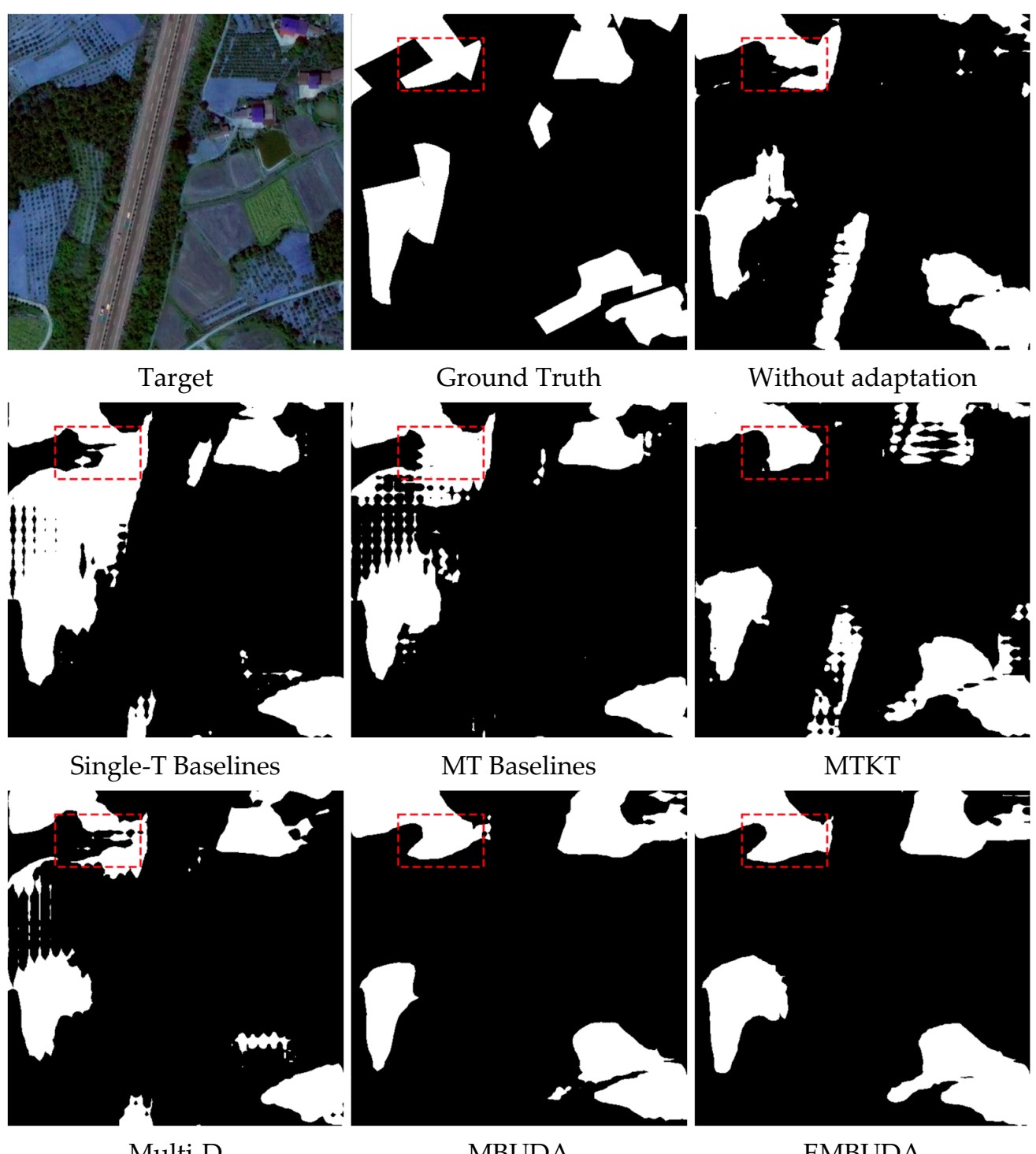

**Figure A8.** Outputs of orchard area segmentation in Dataset XT1 when adapting from Dataset CY to Dataset ZG, Dataset XT1 and Dataset XT2.

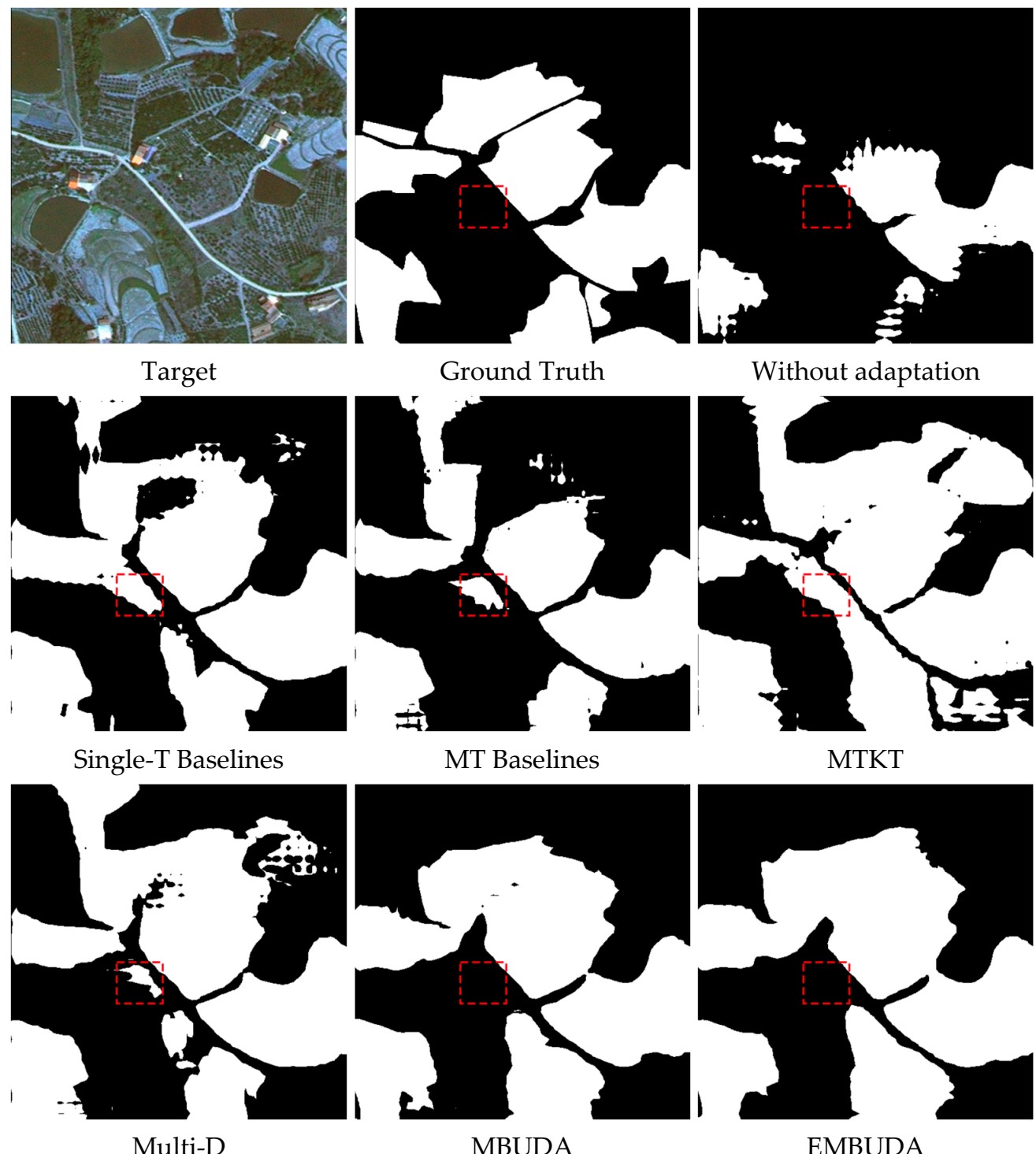

**Figure A9.** Outputs of orchard area segmentation in Dataset XT2 when adapting from Dataset CY to Dataset ZG, Dataset XT1 and Dataset XT2.

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
