# Peer review of "Multibranch Unsupervised Domain Adaptation Network for Cross Multidomain Orchard Area Segmentation"

_remotesensing, doi:10.3390/rs14194915_

Round 1

Reviewer 1 Report

(1) As the statement on line 32, page1, ‘remote sensing data are sensitive to time and region changes, and their acquisition is affected by sensor, shooting angle, and spatial region differences’. Generally, the distribution of training and test set is sensitive to many factors. However, In my view, the appearance of the orchard is more affected by the different types, seasons, and sensors. It suggested making what the domain shift problem focused on in this article clear.

(2) Its advised to add additional detailed legends or descriptions for Figure 2, making it easy to understand. Figure 2 illustrated the transfer from one source domain to two target domains, which means the structure of the model would change according to the number of domains? More importantly, the number of domains is always unknown in advance, especially in practical applications. Does this mean that the method in this paper cannot be used in different scenarios?

(3) It needs to add a more detailed description for the dataset. If the source data in each domain or area has only one image with about 10000*10000, the dataset is too small for a CNN-based method although many data augmentation methods were applied.

(4) On line312, page9, According to the setting with two target domains, four transfer tasks with different domain gaps are selected to prove the validity of the proposed model. How designed the four tasks in the experiment, why not exchange the source and the target domains, such as transfer from dataset XT1 to dataset ZG and dataset CY (XT1ZG + CY).

(5) Figures 5 to 9 as well as Figures A1A9 are difficult to distinguish and compare the extracted results of different methods. Its advised the author to display images separately from extraction results, as the majority of the interpretation of land cover types in remote sensing images. The author can refer to MAP-Net: Multiple Attending Path Neural Network for Building Footprint Extraction From Remote Sensed Imagery or  Cascaded Residual Attention Enhanced Road Extraction from Remote Sensing Images to show the experimental results, which is more friendly for the readers.

Author Response

Thank you for your precious comments and advice. Those comments are all valuable and very helpful for revising and improving our paper, as well as the important guiding significance to our researches. We have studied comments carefully and have made corrections which we hope meet with approval. Please see the attachment for details.

Reviewer 2 Report

The paper addresses an unsupervised learning approach for segmenting orchards considering multiple target domains. The topic is original and presents contributions to the remote sensing community. The experiments are coherent and in hand with previous research. However, I have some additional comments for improving the paper. 

1 – The text must have an English revision. 

2 – The authors mention using UDA models based on adversarial learning. What other types of models exist? What is the advantage of this approach? It might be interesting to mention the main UDA strategies for semantic segmentation (classifier discrepancy, generative, curriculum learning, entropy minimization, and self-training) 

3 - I consider the “Dataset” topic to be part of “Materials and Methods” rather than “Result”. How many samples were generated before the data augmentation? Do the samples overlap or not? 

4 – I suggest that the four transfer tasks with different domain gaps should be described in the methodology. These choices could be justified and better described.  

5 – The full term must precede the acronyms that appear for the first time. Example TP, FN, FP (line 297), RGB (line 169), FADA (line 182), MTKT, EMBUDA (line 308). 

6 – All figures need to improve resolution to facilitate the reading and visualization of the images. Figures must be self-explanatory, inserting the meaning of the acronyms in the legend.  

Figure 1 could be enlarged. Instead of placing the two frames horizontally, they could be placed vertically. 

Figure 3 –The MBUDA framework is impossible to visualize. I see no need to put the MBUDA figure, as it is the same as Figure 2.  

Figure 5 – The legend should better explain the meaning of the colors. What would be the segmentation area? Note that the red color should be described in the legend. 

Figure 6 – I see no need to split Figure 6. Make a single figure and adjust the text layout. For example, table 4 can be placed after figures 5 and 6. 

7 – Table captions should also include the meanings of the acronyms used.  

Tables 3, 4, e 5 – The table captions do not mention that the results refer to the IoU 

8 - The discussion should be improved by comparing the results with other references, highlighting the limitations and advantages of the method, and describing future works. The present discussion should be a subtopic of the results. 

Minor corrections 

Line 15 - “neglect learning and controlling” I suggest “neglect to learn and control”  

Line 21 - “to further reduce the distribution gaps” I suggest “to reduce the distribution gaps further” 

Line 21 - The text presents both forms “adaptation-enhanced learning strategy” (Line 309) and “adaptation enhanced learning strategy” (Lines 21, 90, 93, 243, 247, 263, 409). I suggest standardizing the term using the hyphen.  

Line 30 - “estimation and industrial” I suggest “estimation, and industrial” 

Line 34 - “[6] and other” I suggest “[6], and other” 

Lines 41-44 – Redundant phrase. I suggest “Notably, remote sensing technology can easily acquire a wide variety of data, which are sensitive to time and region changes, and their acquisition is affected by the sensor, shooting angle, and spatial region differences [12, 13].” 

Line 51-53 – “This paper studies UDA models based on adversarial learning [14, 15], which …through adversarial learning “Redundant phrase. “This paper studies UDA models based on adversarial learning [14, 15], which reduce domain gaps and learn domain-invariant features.” 

Lines 56-57 - “applications; however,” I suggest “applications. However,” 

Line 76 - “how to reduce” I suggest “how reducing” 

Lines 119-120 - “The main idea in self-supervision” I suggest “The main idea of self-supervision” 

Line 101 - “methods and self-supervision methods” I suggest “methods, and self-supervision methods”  

Line 106 - “some researchers started to focus” I suggest “some researchers focused” 

Line 131 - “constraints and pseudo-label constraint” I suggest “constraints, and pseudo-label constraint” 

Lines 133-134 - “aerial images classification” I suggest “aerial image classification” 

Lines 185-186 - “If a multipart figure is made up of multiple figure types (one part is line art, and another is grayscale or color), the figure should meet the stricter guidelines.” I suggest “The figure should meet the stricter guidelines if a multipart figure is made up of multiple figure types (one part is line art, and another is grayscale or color).” 

Line 201 - “extractor and the” I suggest “extractor, and the” 

Line 261 - “in discussion.” I suggest “in the discussion.” 

Line 265 - “details of datasets” I suggest “details of the datasets” 

Line 281 - “datasets that have significant” I suggest “datasets with significant” 

Line 292 - “The ratio of high-confidence part and low-confidence part” I suggest “The ratio of high-confidence and low-confidence parts” 

Lines 296-297 “pixels respectively.” I suggest “pixels, respectively.” 

Line 320 - “23.90% and 15.36%” I suggest “23.90%, and 15.36%” 

Line 320 - “Dataset XT1 and Dataset XT2,” I suggest “Dataset XT1, and Dataset XT2,” 

Lines 325-326 “2.21% and 1.84%” I suggest “2.21%, and 1.84%” 

Line 329 - “6.72% and 4.62%” I suggest “6.72%, and 4.62%” 

Line 342 - “6.29% and 3.49%” I suggest “6.29%, and 3.49%” 

Line 379 - “has superior” I suggest “has a superior” 

Author Response

Many thanks for your positive comments on the paper. And we really appreciate for your careful reading on our manuscript and the valuable suggestions. We have studied comments carefully and tried our best to revise our manuscript and made great changes in the manuscript. And we hope that the corrections will meet with approval. Please see the attachment for details.

Reviewer 3 Report

The research manuscript entitled “Multibranch unsupervised domain adaptation network for cross multidomain orchard area segmentation,” is well-organized along with adequate details and descriptions. The results shown in the manuscript are promising. However, there are few comments that needs to be addressed before it can be recommended for publication in the journal of Remote Sensing, MDPI.

1. The main innovation of this study is not clearly stated throughout the manuscript. The reviewer suggests that the authors spend a great amount of effort to better clarify the novelty of their proposed technique. In the introduction section, many recent and relevant contributions close to the authors’ research work are not reviewed. The authors should consider and mention some more recent and relevant references to the literature review to create a proper perspective of the existing state-of-the-art in the area of interest.

2.   â€‹In addition to Table 2 reporting four datasets for domain adaptation, that would better investigate or at least discuss the effects of different training dataset sets on the capability of the presented method, as it is a very well-known issue for several Machine Learning techniques. 

3.   Some figures illustrated in this manuscript, (e.g., Figures 1 and 2) look blurred; hence, the authors are suggested to improve their quality and legibility.

4.   There are some English typos throughout the body of manuscript. The authors are needed to thoroughly revise the paper in terms of faulty sentence structure, sentence fragment, punctuation errors, etc. and correct the English grammar mistakes.

5. More details and references concerning the Multi-target and Multi-branch unsupervised domain adaptation techniques proposed in this manuscript are needed. How exactly is this method superior to the other existing state of the art techniques?

6. The authors should provide information on the computational tool used to implement the framework presented. (e.g., Matlab, Mathematica, or standalone computer code).

7. The authors have mentioned throughout the manuscript that an adaptation enhanced learning strategy was designed and proposed to reduce the target-target domain gaps and to enhance the adaptation effect; however, not enough explanation is provided concerning this strategy from technical point of view. The authors are suggested to explain it clearly throughout the manuscript.

8. More details and explanations are required regarding Appendix A. Some equations and formulations in the text need references and/or mathematical demonstrations.

Author Response

Thank you very much for your positive comments on our manuscript. And we really appreciate for your careful reading on our paper and the valuable suggestions. We have studied comments carefully and tried our best to revise and improve our manuscript and made great changes in the paper according to your good comments. Please see the attachment for details.

Round 2

Reviewer 1 Report

There is no comment.